# On the Relationship Between Variational Inference and Auto-Associative Memory

**Louis Annabi**
ETIS UMR 8051
CY Cergy Paris Université, ENSEA, CNRS
Cergy, France
louis.annabi@gmail.com

**Alexandre Pitti**
ETIS UMR 8051
CY Cergy Paris Université, ENSEA, CNRS
Cergy, France
alexandre.pitti@ensea.fr

**Mathias Quoy**
ETIS UMR 8051
CY Cergy Paris Université, ENSEA, CNRS
Cergy, France
mathias.quoy@ensea.fr

## Abstract

In this article, we propose a variational inference formulation of auto-associative memories, allowing us to combine perceptual inference and memory retrieval into the same mathematical framework. In this formulation, the prior probability distribution onto latent representations is made memory dependent, thus pulling the inference process towards previously stored representations. We then study how different neural network approaches to variational inference can be applied in this framework. We compare methods relying on amortized inference such as Variational Auto Encoders and methods relying on iterative inference such as Predictive Coding and suggest combining both approaches to design new auto-associative memory models. We evaluate the obtained algorithms on the CIFAR10 and CLEVR image datasets and compare them with other associative memory models such as Hopfield Networks, End-to-End Memory Networks and Neural Turing Machines.

## 1 Introduction

In the recent years, many methods have been proposed in order to augment deep learning models with long-term memories. These models provide writing and reading mechanisms allowing to store and retrieve certain patterns inside a memory. Among these models, associative memories are memories that perform content-based addressing, meaning that they retrieve stored patterns based on an approximate or incomplete version of those.

While many models consider one-step reading mechanisms to retrieve patterns from the memory, biological memory retrieval is an iterative process that can take a variable amount of time. Taking this into consideration, Hopfield networks [15] have been proposed as an Associative Memory (AM) implemented by a Recurrent Neural Network (RNN). Hopfield networks perform memory retrieval as an iterative process where stored patterns constitute local attractors of the RNN dynamics.

In this work, we show how AM models such as Hopfield networks can be formulated as variational inference methods using a memory dependent probabilistic model of the observed data. In variational inference algorithms such as Variational Auto-Encoders (VAE) [19, 32] and Predictive Coding (PC) [31, 12], the representation inferred from the observed data tries to minimize an energy function

36th Conference on Neural Information Processing Systems (NeurIPS 2022).

based on the probabilistic model $p(\boldsymbol{x})$ generating the observed data. In the Free Energy Principle (FEP) literature [10, 11], this function is called Variational Free-Energy (VFE), and is equivalent to the negative Evidence Lower Bound (ELBO) more often used in machine learning. The probabilistic model $p(\boldsymbol{x})$ can be decomposed according to a prior probability distribution over representations $p(\boldsymbol{z})$, and a likelihood $p(\boldsymbol{x}|\boldsymbol{z})$ describing the probability of observed data based on the representation. We propose to make $p(\boldsymbol{z})$ depend on the patterns stored in memory, which allows us to derive a new expression of the VFE. The obtained energy function has minima that should be close to one of the stored patterns while properly encoding the observed input. As such, inference based on this energy function can be seen both as perceptual inference and memory retrieval. While using memory-dependent generative models or Gaussian Mixture Models (GMM) with VAEs [9, 3, 23] are not novel ideas, the PC approach has never been to this problem.

The paper is organized as follows: in section 2, we provide a deeper presentation of the concepts connected to our work and review related approaches. In section 3 we present an overview of the FEP mathematical framework and derive an expression of the VFE depending on the patterns stored in memory. In section 4 we design several AM models minimizing this energy function. In section 5, we evaluate the obtained algorithms on the task of memory retrieval on two image datasets, and compare their performance with other AM models.

The contributions brought by this work are the following:

- We design four AM models based on a variational inference formulation of memory retrieval.

- We draw a connection between PC and the modern continuous Hopfield network [30].

## 2 Related work

### 2.1 Auto-associative memory

Several approaches to the long-term storage of information in artificial neural networks have been proposed. Memory networks [38], and End-to-End Memory Networks (MemN2N) [34] propose to store information in a memory matrix that can be addressed using attention coefficients computed based on the content of different memory locations. The Neural Turing Machine (NTM) [13] and the differentiable neural computer [14] combine this content-based addressing using attention mechanisms with a location-based addressing allowing more computer-like memory accesses.

Standing out from these approaches that consider a feedforward reading mechanism, Hopfield networks [15] and continuous Hopfield networks [16] instead store patterns as local attractors of an RNN. These memories can thus be addressed by initializing the RNN state with the input pattern, and retrieving the stored attractor after convergence. In order to improve the memory capacity, modern Hopfield networks [22, 21, 8] propose several variants of the energy function using polynomial or exponential interactions. Extending these models to the continuous case, [30] proposed the Modern Continuous Hopfield Network (MCHN) with update rules implementing self attention, that they relate to the transformer model [36]. In [26], the authors introduce a general Hopfield network framework where the update rules are built using three components: a similarity function, a separation function, and a projection function.

It has been shown that overparameterized auto-encoders also implement AM [28, 33]. These methods embed the stored patterns as attractors through training, and retrieval is performed by iterating over the auto-encoding loop. In contrast, our methods allow one-shot writing: a new pattern $\boldsymbol{x}$ can be inserted in memory simply by computing its representation and adding it to the memory matrix $\boldsymbol{M}$.

Another line of research takes inspiration from the Sparse Distributed Memory model [18], building connections with attention mechanisms [4] and with the variational inference framework [40, 41, 29]. In particular, the Kanerva Machine [40, 41, 29] is similar to the models we build in many aspects: they use an iterative reading mechanism, a memory-dependent prior on the representation, all within the variational inference framework. Though, the PC toolbox provides methods that set the proposed models apart from these approaches, as detailed in the methods section.

## 2.2 Predictive Coding

PC is a theory of brain function [31, 7] extending the idea that neural representations emerge as part of an inference process of the causes of sensory observations, as already suggested by Helmholtz in 1867 [37]. The FEP [10, 11] provides a principled derivation of PC networks based on the minimization of variational free-energy (VFE), a quantity equivalent to the negative evidence lower bound (ELBO) used in variational Bayesian methods, and defined as:

$$F(\boldsymbol{x}) = \int_z \log\left(\frac{q(\boldsymbol{z})}{p(\boldsymbol{x}, \boldsymbol{z})}\right) q(\boldsymbol{z}) d\boldsymbol{z} \tag{1}$$

where $p(\boldsymbol{x}, \boldsymbol{z})$ denotes the generative model and $q(\boldsymbol{z})$ denotes the approximate posterior (also called recognition density) on $\boldsymbol{z}$. VAEs [19, 32] are a well-known method applying the idea of VFE minimization (equivalently ELBO maximization) to neural networks. In VAEs, this quantity is only optimized during the model training, and perceptual inference is performed as a simple forward pass through the encoder.

In contrast, neural network models based on PC intertwine the prediction (decoder) and inference (encoder) mechanisms within a single hierarchical recurrent architecture comprising a population of representation neurons and a population of prediction error neurons at each layer. Perceptual inference is then an iterative mechanism supported by the dynamics of this RNN. Given an observed input $\boldsymbol{x}$, the representation $\boldsymbol{z}$ is updated at each time step based on a bottom-up signal pushing $\boldsymbol{z}$ towards values that minimize the reconstruction error, and on a top-down signal pushing $\boldsymbol{z}$ towards values that maximize its prior probability $p(\boldsymbol{z})$. While this prior probability on $\boldsymbol{z}$ is often ignored, we show that it can bring auto-associative capacities to the PC network, and we draw a connection between the obtained mechanism and the modern continuous Hopfield network proposed in [30].

While both VAEs and PC networks output an estimation of the approximate posterior $q(\boldsymbol{z})$, they differ in the computational mechanisms used for inference: VAEs perform amortized inference via a forward pass through an encoder, while PC networks perform an iterative inference implemented by its recurrent dynamics. Some works have suggested combining both approaches: the iterative inference can be initialized using the estimation provided by the amortized inference method [2, 35].

A previous work [1] explored the use PC techniques together with GMM-based prior probability distributions $p(\boldsymbol{z})$. However, the obtained models were not used to perform memory retrieval.

## 3 Memory-based Variational Inference

### 3.1 Framework definition

Our framework is based on a generative model $p(\boldsymbol{x}, \boldsymbol{z})$ that can be factored into a prior probability $p(\boldsymbol{z})$ and a likelihood $p(\boldsymbol{x}|\boldsymbol{z})$. The prior probability on $\boldsymbol{z}$ is defined as a memory dependent distribution:

$$p(\boldsymbol{z}) = p(\boldsymbol{z}; \boldsymbol{M}) \tag{2}$$

where the vectors $\boldsymbol{M}_k$ (the columns of $\boldsymbol{M}$) constitute a repertoire of stored representations.

The probability $p(\boldsymbol{x}|\boldsymbol{z})$ is defined as a hierarchical generative model featuring several intermediate variables $\{\boldsymbol{h}_1, \cdots, \boldsymbol{h}_{L-1}\}$, where $L$ denotes the number of layers. By extension we use the notations $\boldsymbol{h}_0 = \boldsymbol{x}$ and $\boldsymbol{h}_L = \boldsymbol{z}$. We assume that the generative model is a cascade of multivariate Gaussians:

$$p(\boldsymbol{h}_l|\boldsymbol{h}_{l+1}) = \mathcal{N}(\boldsymbol{h}_l; \boldsymbol{f}_{\boldsymbol{\theta}}^l(\boldsymbol{h}_{l+1}), \mathbb{I}) \tag{3}$$

where $\mathbb{I}$ is the covariance matrix of the Gaussians and is uniform across all layers, and $\boldsymbol{f}_{\boldsymbol{\theta}}^l$ are functions (typically neural network layers) parameterized by $\boldsymbol{\theta}$. Note that this can be adapted to arbitrary computation graphs by replacing $\boldsymbol{h}_{l+1}$ by the set of parent nodes of $\boldsymbol{h}_l$ for each node. For simplicity, we assume in the following derivations that each node only has one parent. In our experiments, the functions $\boldsymbol{f}_{\boldsymbol{\theta}}^l$ correspond to the different layers of a Convolutional Neural Network (CNN) on the CIFAR10 dataset, and of a MONet [6] decoder on the CLEVR dataset. We denote by $\boldsymbol{f}_{\boldsymbol{\theta}}$ the composition of all layers $\boldsymbol{f}_{\boldsymbol{\theta}}^l$.

We can derive the VFE corresponding to the described generative model. The FEP formulation of PC uses different approximations that allow us to greatly simplify the expression of the VFE introduced

in equation (1). We refer to appendix A for detailed derivations and simply provide the simplified expression:

$$F(\boldsymbol{x}, \hat{\boldsymbol{h}_1}, \ldots, \hat{\boldsymbol{h}_{L-1}}, \hat{\boldsymbol{z}}) = \sum_{l=0}^{L-1} \frac{1}{2} \|\hat{\boldsymbol{h}_l} - \boldsymbol{f}_{\boldsymbol{\theta}}^l(\hat{\boldsymbol{h}_{l+1}})\|^2 - \log p(\hat{\boldsymbol{z}}; \boldsymbol{M}) + C \qquad (4)$$

where C is a quantity independent from $\{\boldsymbol{x}, \hat{\boldsymbol{h}_1}, \ldots, \hat{\boldsymbol{h}_{L-1}}, \hat{\boldsymbol{z}}\}$. The vectors $\{\hat{\boldsymbol{h}_1}, \ldots, \hat{\boldsymbol{h}_{L-1}}, \hat{\boldsymbol{z}}\}$ correspond to the means of the approximate posterior $q(\boldsymbol{h}_1, \ldots, \boldsymbol{h}_{L-1}, \boldsymbol{z}) = q(\boldsymbol{z}) \prod_{i=1}^{L-1} q(\boldsymbol{h_i})$.

The term corresponding to the prior probability is often omitted, which can be justified as being equivalent to having no prior preferences over different values of $\boldsymbol{z}$. Key to our method is the idea that this prior can pull the inference process towards values of $\boldsymbol{z}$ previously stored in the memory $\boldsymbol{M}$, turning the patterns $\boldsymbol{M}_k$ into attractors of the PC network. As such, a suitable distribution would be one that associates high probabilities for the patterns stored in memory.

### 3.2 Classification of related methods in this framework

Here, we show that we can formulate Modern Continuous Hopfield Networks (MCHN) as PC networks derived from this expression of the VFE under some conditions:

- There is no representation component: $L = 0$.
- The PC network is initialized with $\hat{\boldsymbol{z}} = \boldsymbol{x}$.
- The prior distribution $p(\boldsymbol{z}; \boldsymbol{M})$ is defined as:

$$p_{MCHN}(\boldsymbol{z}; \boldsymbol{M}) = \sum_{k=1}^{N} \pi_k \mathcal{N}\big(\boldsymbol{z}; \boldsymbol{M}_k, \beta^{-1}\mathbb{I}\big) \qquad (5)$$

$$\text{with } \forall k, \pi_k = \frac{\exp\{\frac{\beta}{2} \boldsymbol{M}_k^{\mathsf{T}} \cdot \boldsymbol{M}_k\}}{\sum_{k'=1}^{N} \exp\{\frac{\beta}{2} \boldsymbol{M}_{k'}^{\mathsf{T}} \cdot \boldsymbol{M}_{k'}\}} \qquad (6)$$

This prior distribution is a Gaussian Mixture Model (GMM) with mixture means corresponding to the $N$ stored patterns and mixing coefficients $\pi_k$ depending on the patterns' Euclidean norm. It is parameterized by a coefficient $\beta > 0$. Based on these assumptions, we can derive the following expression for the VFE:

$$F(\hat{\boldsymbol{z}}) = \frac{\beta}{2} \hat{\boldsymbol{z}}^{\mathsf{T}} \cdot \hat{\boldsymbol{z}} - \log \sum_{k=1}^{N} \exp\{\beta \hat{\boldsymbol{z}}^{\mathsf{T}} \cdot \boldsymbol{M}_k\} + C \qquad (7)$$

which is, up to a constant $C$ and a factor $\beta$, equivalent to the MCHN energy function proposed in [30]. The complete derivations of this expression are provided in appendix B. We can note that VFE does not depend on the input $\boldsymbol{x}$. The input is not part of the energy function, but serves as an initial estimate of the approximate posterior mean $\hat{\boldsymbol{z}}$. According to the FEP formulation of PC, applying gradient descent on the VFE with regard to the approximate posterior mean $\hat{\boldsymbol{z}}$ yields the update rules of the PC network. Using our expression of the VFE, we obtain:

$$\hat{\boldsymbol{z}} \leftarrow \text{softmax}(\beta \hat{\boldsymbol{z}} \cdot \boldsymbol{M}) \boldsymbol{M}^{\mathsf{T}} \qquad (8)$$

This equation is identical to the update rule of MCHN (detailed derivations are provided in appendix). We have shown that our framework can be related to MCHNs, but variational inference methods for perceptual inference can also be retrieved in this framework by simply using a neutral prior.

For VAEs and variants, this prior probability distribution is exploited during learning, but the amortized inference cannot be dynamically adapted to this distribution. Therefore, to take into account a change in this prior, we would need to retrain the encoder. In the derivations of PC networks [12, 5], the term $p(\boldsymbol{z})$ is often ignored, which is equivalent to having a flat prior distribution. However,

Table 1: Summary of the related models and proposed models.

| Model | Representation | Associative memory |
|---|---|---|
| MCHN [30] | ✗ | Iterative |
| VAE [19] | Amortized | ✗ |
| PC [31] | Iterative | ✗ |
| BP [27] | Iterative | ✗ |
| HPC [35] | Amortized and iterative | ✗ |
| GMVAE [9] | Amortized | Amortized |
| Overparameterized VAE [28] | Amortized | Iterative |
| Kanerva Machines [40] | Amortized | Iterative |
| VAE-PC-GMM (Ours) | Amortized and iterative | Iterative |
| VAE-BP-GMM (Ours) | Amortized and iterative | Iterative |
| VAE-GMM (Ours) | Amortized | Iterative |

theoretically the iterative inference could take into account a prior distribution, and dynamically adapt to changes in this distribution, without retraining the model.

In the family of iterative inference algorithms, we can also mention the methods based on backpropagation (BP) to estimate the representation. Instead of using neural computations to simulate the gradient descent on the energy function as done in PC, these methods directly use BP to optimize the representation $\hat{z}$. For instance [27] minimize prediction error using BPTT to adjust a latent variable in an RNN. This method is in fact very similar to PC, since it has been shown that under some conditions, PC approximates the update rules entailed by BP applied on the reconstruction error [39, 25]. We provide in table 1 a simple classification of these approaches depending on whether they use a representation and memory component, and whether the inference mechanism is amortized or iterative.

# 4 Methods

In this section, we present the AM models that we have designed using this framework. All these models are based on a pre-trained VAE that is used to provide initial estimates $\tilde{z}$ and predictions of $x$ based on $\tilde{z}$. The proposed methods are represented in figure 1.

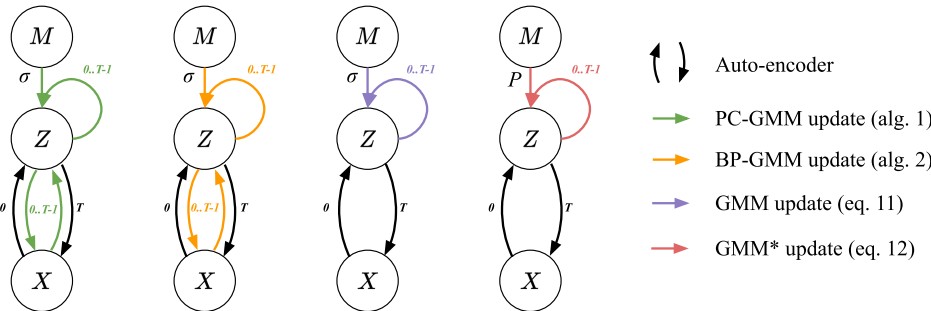

Figure 1: Representation of the memory retrieval process in the different proposed models. From left to right: VAE-PC-GMM (4.2), VAE-BP-GMM (4.3), VAE-GMM (4.4) and VAE-GMM* (4.5).

## 4.1 Probabilistic model

The prior probability over the representation $z$ is crucial to obtain a model that infers representations close to stored patterns. We have shown that MCHN can be obtained starting from a GMM prior biased towards stored patterns of large Euclidean norm. Since there is no intuitive justification for this bias, we choose to instead start with a balanced GMM prior distribution:

$$p(\boldsymbol{z}; \boldsymbol{M}) = \sum_{k=1}^{N} \frac{1}{N} \mathcal{N}(\boldsymbol{z}; \boldsymbol{M}_k, \Sigma) \tag{9}$$

where $\Sigma$ denotes the covariance matrix, uniform across the $N$ mixtures. In the following models, we consider the simpler case where $\boldsymbol{\Sigma} = \sigma^2 \mathbb{I}$, except for the last proposed model (section 4.5) where the precision matrix $\boldsymbol{P} = \boldsymbol{\Sigma}^{-1}$ is trained.

## 4.2 PC based inference (VAE-PC-GMM)

Following the FEP formulation of PC, we can derive the gradient of $F$ (from eq. 4) according to each layer's representation $\hat{\boldsymbol{h}}_l$. As such, performing gradient descent on $F$ yields a system of update rules on each representation, that can be interpreted as RNN dynamics. This PC network comprises at each layer two populations of neurons, one encoding the layer's representation, and one encoding the layer prediction error $\boldsymbol{\epsilon}_l = \hat{\boldsymbol{h}}_l - \boldsymbol{f}_{\boldsymbol{\theta}}^l(\hat{\boldsymbol{h}_{l+1}})$.

Detailed derivations of this model are provided in appendix C, where the forward pass through the model is given in algorithm 1. Since the RNN dynamics implement a gradient descent, the initialization of this network is responsible for the local minimum to which it converges. We label VAE-PC-GMM the version of this algorithm where $\hat{\boldsymbol{z}}$ is initialized using the result of the amortized inference via the encoder of the VAE.

## 4.3 BP based inference (VAE-BP-GMM)

In this second algorithm, we instead use BP as the iterative inference mechanism used to optimize $\hat{\boldsymbol{z}}$. BP minimizes the following loss function:

$$\mathcal{L}(\boldsymbol{x}, \hat{\boldsymbol{z}}) = \|\boldsymbol{f}_{\boldsymbol{\theta}}(\hat{\boldsymbol{z}}) - \boldsymbol{x}\|^2 - \gamma \log p(\hat{\boldsymbol{z}}; \boldsymbol{M}) \tag{10}$$

where $\gamma$ is an hyperparameter weighting the influence of bottom-up and top-down mechanisms, and $\boldsymbol{f}_{\boldsymbol{\theta}}$ denotes the decoder of the VAE. This gradient descent is parameterized by a learning rate $\lambda$. Once again, we can initialize the gradient descent using the estimate obtained with the encoder of the VAE. Derivations of this model are provided in appendix C, where the forward pass through the model is given in algorithm 2.

Investigating the relationship between PC and BP based inference, we have found that PC networks needed a larger number of iterations to convey information from the reconstruction error, and that this number of iterations was exponential with regard to the depth of the decoder. For this reason, experiments with the VAE-PC-GMM model were prohibitively slow, and we only conducted experiments with the remaining models.

## 4.4 Restricting the iterative inference to the memory component (VAE-GMM)

Backpropagating the reconstruction error might not be necessary when the estimate provided by the VAE encoder already conveys enough information from the observed input $\boldsymbol{x}$. As such, we also experiment with a simpler version of the previous model where iterative inference of $\hat{\boldsymbol{z}}$ only considers the top-down update rule coming from the memory. In this simpler version, the update rule for $\hat{\boldsymbol{z}}$ provided in algorithm 1 becomes (proof in appendix C):

$$\hat{\boldsymbol{z}} \leftarrow \text{softmax}\left(-\frac{\|\hat{\boldsymbol{z}} - \boldsymbol{M}\|_2^2}{2\sigma^2}\right) \cdot \boldsymbol{M}^{\mathsf{T}} \tag{11}$$

This update mechanism is computationally lighter than the BP-based inference model, and has the advantage of being differentiable.

## 4.5 Training the precision matrix (VAE-GMM*)

For this last model, we suggest optimizing the precision coefficients $\boldsymbol{P} = \boldsymbol{\Sigma}^{-1}$ of the GMM. These coefficients condition the shape of the Gaussian mixtures. Some memory retrieval tasks might need

to ignore partial information from the observed input $x$ and as such could benefit from such an adaptation. In our experiments, we design such tasks on the CLEVR dataset, where the objective is to retrieve scenes using as input a shifted image of the same scene, or an image of the same scene where the colors of the objects have been modified. To properly accomplish these two tasks, the AM model needs to learn to give less importance to position information in the first case, and to color information in the second case.

The model starts from the estimate $\hat{z}$ inferred by the VAE, and uses the following update rule:

$$\hat{z} \leftarrow \text{softmax}\Big( -\frac{1}{2}(\hat{z} - M)^{\intercal} \cdot P \cdot (\hat{z} - M) \Big) \cdot M^{\intercal} \tag{12}$$

During training, $P$ is optimized using BP in order to reduce the mean squared error between the inferred representation $\hat{z}$ and the correct memory pattern $M_k^*$. During evaluation, the values of $P$ are fixed.

## 5 Experiments

In this section, we present the experiments performed to evaluate the proposed models. In all experiments, we measure performance using the percentage of properly retrieved patterns from associative memories containing $N = 100$ patterns. We consider that a pattern is properly retrieved if the distance between the inferred representation $\hat{z}$ and the correct memory pattern $M_{k*}$ is lower than a threshold value chosen manually. All the implementation details (benchmark models, hyperparameters, training, dataset splits) are provided in appendix D.

### 5.1 Datasets

We evaluate the proposed models on two image datasets: CIFAR10 and CLEVR. CIFAR10 [20] (MIT License) consists of $32 \times 32$ RGB images of 10 classes. We pretrain a VAE on the training set using a convolutional architecture, with a latent space of dimension $d = 16$.

The CLEVR dataset [17] (CC BY 4.0 License) consists of $64 \times 64$ RGB images of 3D scenes composed of simple 3D objects. We use a pretrained MONet [6] model as the VAE, with a latent space of dimension $d = 4 \times 16$. The MONet encoder infers a set representation given an input scene image. In our experiments, we flatten this set representation, which can lead to wrong measures of similarity between two representations. Indeed, a permuted set representation still encodes the same information, but once flattened the permuted and original representations might have a low similarity score. To limit this issue, we have capped the number of objects present in the scene images to 3, and capped the size of the set representation to 4. Examples of images from the CLEVR dataset along with possible transformations are presented in figure 2.

### 5.2 Benchmark models

We compare our AM models with Modern Continuous Hopfield Networks (MCHN), Neural Turing Machines (NTM) and End-to-End Memory Networks (MemN2N). For fair comparison, in all benchmark models, the memory is initialized with the stored patterns (we do not use the writing mechanisms of NTMs). We also experiment with these models in the representation space, where the memory is initialized with representations of the stored images, and the networks receive as input the representation of the corrupted image. We indicate this modification with the VAE prefix.

The NTM and MemN2N models are trained with BP using as loss function the mean squared error between the predicted output $\hat{x}$ (respectively $\hat{z}$ when using the VAE) and the correct memory pattern $M_k^*$. For a fair comparison, this training is performed with clean inputs $x = M_k^*$, to avoid giving any prior information about the type of transformation the inputs might be corrupted with during evaluation. Finally, we also experiment with a version of the MCHN using a balanced GMM as starting prior distribution, that we denote GMM in our experiments. Equivalently, this is the version of our VAE-GMM algorithm without a representation component.

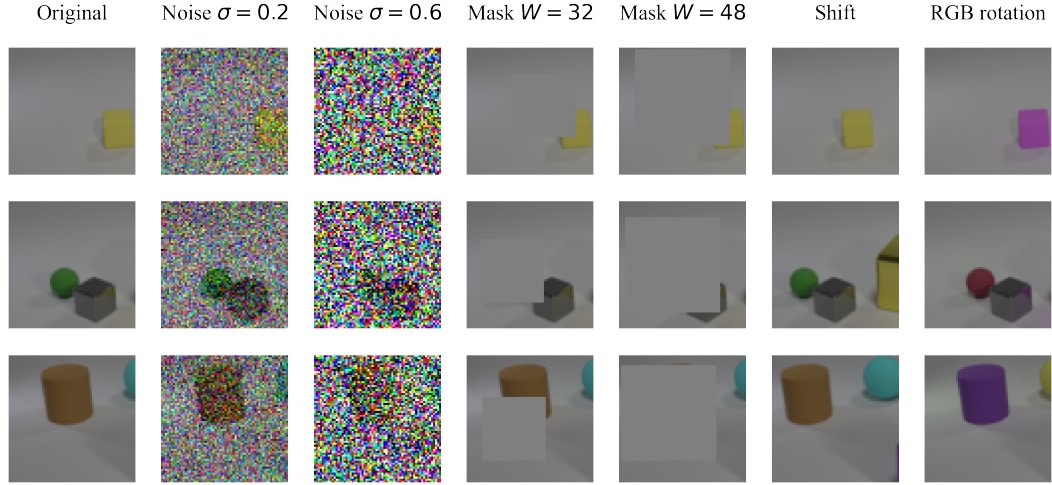

Figure 2: Examples of original, corrupted, and transformed images from the CLEVR dataset.

## 5.3 Corrupted inputs

First, we evaluate whether the associative memory models are able to properly retrieve stored patterns when given corrupted versions of these patterns. We apply different level of noise to the images ($\sigma \in \{0.2, 0.4, 0.6, \cdots, 2.0\}$), as well as different sizes of masks (16 and 24 pixels wide on the CIFAR10 dataset, and 32 and 48 pixels wide on the CLEVR dataset). The results of these experiments are reported in table 2 and 3. More figures and examples are provided in appendix E.

Table 2: Memory retrieval with corrupted inputs on the CIFAR10 dataset. Mean percentages of successful retrieval and standard deviations are reported.

| Input | Clean input | Noise $\sigma$=0.2 | Noise $\sigma$=0.6 | Mask W=16 | Mask W=24 |
|---|---|---|---|---|---|
| GMM | **100 ± 0.0** | **100 ± 0.0** | 94.3 ± 6.6 | **99.7 ± 1.3** | 48.7 ± 12.7 |
| Hopfield | 3.0 ± 3.9 | 3.9 ± 4.0 | 1.5 ± 2.4 | 2.3 ± 3.6 | 1.6 ± 2.6 |
| NTM | 99.3 ± 2.4 | 98.9 ± 2.7 | 91.6 ± 15.0 | 94.7 ± 8.0 | **62.7 ± 20.9** |
| MemN2N | 81.1 ± 11.3 | 77.9 ± 11.1 | 65.8 ± 13.4 | 63.2 ± 13.9 | 32.3 ± 11.5 |
| VAE-BP-GMM | **100 ± 0.0** | **100 ± 0.0** | 81.1 ± 8.9 | 98.1 ± 3.3 | 39.5 ± 10.7 |
| VAE-GMM | **100 ± 0.0** | **100 ± 0.0** | 86.3 ± 9.1 | 98.3 ± 2.8 | 39.5 ± 11.8 |
| VAE-Hopfield | 92.5 ± 6.2 | 91.7 ± 6.0 | 79.6 ± 10.3 | 82.5 ± 8.1 | 52.5 ± 10.1 |
| VAE-NTM | **100 ± 0.0** | **100 ± 0.0** | **98.6 ± 2.3** | 99.5 ± 1.5 | 58.7 ± 10.2 |
| VAE-MemN2N | 95.0 ± 4.7 | 92.6 ± 6.1 | 82.2 ± 8.7 | 82.2 ± 8.0 | 44.4 ± 11.7 |

Table 3: Memory retrieval with corrupted inputs on the CLEVR dataset. Mean percentages of successful retrieval and standard deviations are reported.

| Input | Clean input | Noise $\sigma$=0.2 | Noise $\sigma$=0.6 | Mask W=32 | Mask W=48 |
|---|---|---|---|---|---|
| GMM | **100 ± 0.0** | **100 ± 0.0** | **73.0 ± 10.1** | 69.3 ± 8.6 | 30.3 ± 10.9 |
| Hopfield | 1.3 ± 2.3 | 1.3 ± 2.8 | 1.0 ± 2.5 | 0.8 ± 1.8 | 1.5 ± 2.4 |
| NTM | 82.1 ± 13.1 | 84.5 ± 11.9 | 47.8 ± 16.8 | 56.4 ± 13.2 | 17.6 ± 9.7 |
| MemN2N | 17 ± 11.3 | 16 ± 12.3 | 14.5 ± 10.5 | 9 ± 11.2 | 5.5 ± 6.0 |
| VAE-BP-GMM | **100 ± 0.0** | 49.5 ± 16.0 | 2.0 ± 2.5 | 69.2 ± 10.2 | 36.5 ± 5.8 |
| VAE-GMM | **100 ± 0.0** | 46.3 ± 11.8 | 4.4 ± 4.4 | 68.9 ± 9.7 | 31.5 ± 11.6 |
| VAE-Hopfield | 94.2 ± 4.9 | 34.5 ± 11.4 | 2.7 ± 4.0 | 64.4 ± 11.5 | 32.7 ± 9.9 |
| VAE-NTM | 99.9 ± 0.7 | 40.8 ± 10.0 | 3.4 ± 4.3 | **75.0 ± 10.2** | **38.3 ± 11.2** |
| VAE-MemN2N | 97.9 ± 3.1 | 31.7 ± 11.4 | 2.4 ± 3.1 | 66.1 ± 10.4 | 34.2 ± 10.1 |

We can observe that the two models using dot-product attention, MCHN (Hopfield in the tables) and MemN2N, perform poorly compared to other methods, not even reaching 100% of correctly retrieved patterns when presented with clean inputs. The NTM model, using cosine similarity based attention, works better and often outperforms the proposed models. Among the proposed models, we can see that the VAE-BP-GMM does not perform significantly better than the simpler VAE-GMM model.

An important observation from these results is that in most scenarios, the best performing methods do not use a representation component. In particular, the MONet model seems very sensitive to noise. However, our intuition is that memory retrieval on the representation level should be more powerful if the transformation applied on the observed inputs $x$ has a limited effect on the representations.

## 5.4   Scene transformations

To verify this hypothesis, we perform two additional experiments on the CLEVR dataset. If we assume that the representations provided by MONet encode object positions, shapes and colors, then an AM model working on the representation level could perform well when we apply transformations on these specific object features. We propose two such scenarios. In the first scenario, we perform an RGB rotation on the images, which results in identical scenes with the exception of the object colors. In the second scenario, we take a shifted crop of the original CLEVR image, which can simulate a shifted point of view of the same scene. Examples images are displayed in figure 2.

We measure the success of memory retrieval with these transformed inputs using our models as well the NTM and MemN2N models. We also experiment with the VAE-GMM* model performing precision coefficient training. We expect the model to learn that some information conveyed by the encoder (for instance color in the first scenario) is irrelevant for memory retrieval. For fair comparison, we also experiment with the VAE-NTM and VAE-MemN2N models trained in the two scenarios, that we denote with an asterisk. The results are reported in table 4.

Table 4: Memory retrieval with transformed inputs on the CLEVR dataset. Mean percentages of successful retrieval and standard deviations are reported.

| Input | Color rotation | Shift |
|---|---|---|
| GMM | $76 \pm 12.4$ | $28 \pm 8.5$ |
| NTM | $48.8 \pm 16.9$ | $0.7 \pm 1.9$ |
| MemN2N | $12.6 \pm 11.5$ | $6.4 \pm 8.1$ |
| VAE-GMM | $53.6 \pm 12.0$ | $63.1 \pm 10.2$ |
| VAE-NTM | $54.6 \pm 10.7$ | $61.5 \pm 11.6$ |
| VAE-MemN2N | $41.2 \pm 11.0$ | $56.2 \pm 12.1$ |
| VAE-GMM* | $\mathbf{98.9 \pm 2.5}$ | $93.7 \pm 5.3$ |
| VAE-NTM* | $98.2 \pm 3.2$ | $\mathbf{96.3 \pm 4.4}$ |
| VAE-MemN2N* | $91.6 \pm 6.9$ | $84.2 \pm 7.8$ |

Using a representation component improves the performance of the models in these scenarios. Additionally, training on the retrieval task where the transformations are applied provides almost perfect retrieval scores for the VAE-GMM and VAE-NTM models.

## 6   Discussion

We have proposed a formulation of auto-associative memories based on the FEP and the variational inference framework, where the stored patterns condition the prior probability on the representation $z$. This framework has allowed us to draw a connection between MCHNs and the PC theory, as well as to design several AM models with high retrieval accuracy and robustness.

Combining representation and memory allows to retrieve patterns on the representation level, which decreases the size of the memory store, and performs retrieval based on a similarity measured in the latent space, where we can expect more meaningful features to appear. For instance, we have shown that this allows to recognize visual scenes from a shifted point of view, while pixel-level AM models failed most of the time. Our results also seem to demonstrate that distance based attention as

used in our models (see equation 11) outperforms dot-product based attention as used in MCHNs, although this performance gap could be mitigated with normalization techniques, as hinted by the results obtained with NTMs, that use cosine similarity instead of dot-product.

The proposed PC and BP based models take into account information from the query $x$ at each inference iteration, by directly minimizing an energy function that depends on $x$ (eq. 4 and eq. 10). In contrast, the Kanerva Machine [40, 41] or the VAE-GMM and VAE-GMM* models we proposed only use this information to output an initial estimate $\hat{z}$ that is later optimized in order to minimize an energy function only depending on the memory. According to our experiments, this feature of the PC and BP based models does not seem useful for memory retrieval. However, in other applicative settings, these models could benefit from their ability to weight query pattern information and stored patterns information, performing a form of memory-aided iterative perceptual inference.

The proposed formulation is limited to content-based reading mechanisms. In contrast, models such as NTMs offer a larger variety of addressing schemes as well as writing mechanisms that allow to only store relevant information in the memory. Another limitation of this work is that we did not investigate the memory capacity and convergence properties of the proposed models. Future work should focus on these analyses, and possible improvements of the PC-based inference, for instance using precision weighting, or the "learning to optimize" approach of [24].

# 7   Acknowledgements

This work was funded by the CY Cergy-Paris University Foundation (Facebook grant) and partially by Labex MME-DII, France (ANR11-LBX-0023-01).

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
