## A    General VFE derivation

In this section, we provide the derivations for the VFE expression given in equation 4.

The FEP formulation of PC is based on several assumptions that allow to derive a simpler expression of the VFE. For simplicity, we provide a derivation for the case where $L = 1$, meaning that there is only one layer in the generative model of $\boldsymbol{x}$ based on $\boldsymbol{z}$. At the end of the derivation, we provide the expression in the general case. We start from a different equivalent formulation of the VFE:

$$F(\boldsymbol{x}) = \int_{\boldsymbol{z}} E(\boldsymbol{x}, \boldsymbol{z})q(\boldsymbol{z})d\boldsymbol{z} + \int_{\boldsymbol{z}} \log\big(q(\boldsymbol{z})\big)q(\boldsymbol{z})d\boldsymbol{z} \tag{13}$$

where $E(\boldsymbol{x}, \boldsymbol{z}) = -\log p(\boldsymbol{x}, \boldsymbol{z})$ is called the energy. We assume that the distribution $q(\boldsymbol{z})$ takes a Gaussian form $q(\boldsymbol{z}) = \mathcal{N}(\boldsymbol{z}; \hat{\boldsymbol{z}}, \zeta\mathbb{I})$. The mean of this approximate posterior, $\hat{\boldsymbol{z}}$, corresponds to the inferred representation being optimized by the PC networks. Integrating this definition into the VFE expression, we obtain:

$$F(\boldsymbol{x}) = -\frac{d}{2}\log(2\pi\zeta) - \frac{d}{2} + \int_{\boldsymbol{z}} E(\boldsymbol{x}, \boldsymbol{z})q(\boldsymbol{z})d\boldsymbol{z} \tag{14}$$

We assume that the approximate posterior is tightly shape around its mean $\hat{\boldsymbol{z}}$, allowing us to use the Taylor expansion of $E(\boldsymbol{x}, \boldsymbol{z})$ around this value:

$$E(\boldsymbol{x}, \boldsymbol{z}) \approx E(\boldsymbol{x}, \hat{\boldsymbol{z}}) + \big(\nabla_{\hat{\boldsymbol{z}}} E(\boldsymbol{x}, \hat{\boldsymbol{z}})\big) \cdot (\boldsymbol{z} - \hat{\boldsymbol{z}}) \tag{15}$$

We can now derive an expression of the VFE that depends on $\hat{\boldsymbol{z}}$ and not longer involves integrals:

$$F(\boldsymbol{x}, \hat{\boldsymbol{z}}) \approx E(\boldsymbol{x}, \hat{\boldsymbol{z}}) + \big(\nabla_{\hat{\boldsymbol{z}}} E(\boldsymbol{x}, \hat{\boldsymbol{z}})\big) \cdot \int_{\boldsymbol{z}} (\boldsymbol{z} - \hat{\boldsymbol{z}})q(\boldsymbol{z})d\boldsymbol{z} + C \tag{16}$$

$$\approx E(\boldsymbol{x}, \hat{\boldsymbol{z}}) + C \tag{17}$$

$$\approx -\log p(\boldsymbol{x}|\hat{\boldsymbol{z}}) - \log p(\hat{\boldsymbol{z}}; \boldsymbol{M}) + C \tag{18}$$

where C is a quantity that does not depend on $\boldsymbol{x}$ and $\hat{\boldsymbol{z}}$. Finally, generalizing this expression to $L$ layers, and assuming that each layer in the generative model takes the form of a Gaussian distribution with mean $\boldsymbol{f}_{\boldsymbol{\theta}}^l(\hat{\boldsymbol{h}_{l+1}})$ and variance $\mathbb{I}$, we obtain:

$$\begin{aligned} F(\boldsymbol{x}, \hat{\boldsymbol{h}_1}, \ldots, \hat{\boldsymbol{h}_{L-1}}, \hat{\boldsymbol{z}}) = \sum_{l=0}^{L-1} \frac{1}{2}\|\hat{\boldsymbol{h}_l} - \boldsymbol{f}_{\boldsymbol{\theta}}^l(\hat{\boldsymbol{h}_{l+1}})\|^2 \\ - \log p(\hat{\boldsymbol{z}}; \boldsymbol{M}) \\ + C' \end{aligned} \tag{19}$$

where $C'$ includes other terms independent from $\{\boldsymbol{x}, \hat{\boldsymbol{h}_1}, \cdots, \hat{\boldsymbol{z}}\}$ coming from the derivation of the logarithms of the multivariate Gaussians. More detailed derivations can be found in [5], without the memory dependency, but this has virtually no impact on the derivations.

# B  MCHN derivations

In this section, we provide the derivations for the expression of the VFE and the update rule for MCHNs (equations 7 and 8 in the main text). Using the assumptions listed in section 3.2, we can derive an expression of the VFE that closely resembles the energy function proposed in the MCHN paper [30]:

$$F(\hat{z}) = -\log p(\hat{z}; \boldsymbol{M}) + C \tag{20}$$

$$= -\log \sum_{k=1}^{N} \frac{\exp\{\frac{\beta}{2}\boldsymbol{M}_{\boldsymbol{k}}^{\mathsf{T}} \cdot \boldsymbol{M}_{\boldsymbol{k}}\}}{\sum_{k'=1}^{N} \exp\{\frac{\beta}{2}\boldsymbol{M}_{\boldsymbol{k'}}^{\mathsf{T}} \cdot \boldsymbol{M}_{\boldsymbol{k'}}\}} \frac{1}{\sqrt{2\pi\beta^{-d}}} \exp\{-\frac{\beta}{2}(\hat{z} - \boldsymbol{M}_{\boldsymbol{k}})^{\mathsf{T}} \cdot (\hat{z} - \boldsymbol{M}_{\boldsymbol{k}})\} + C \tag{21}$$

$$= -\log \sum_{k=1}^{N} \exp\{\frac{\beta}{2}\boldsymbol{M}_{\boldsymbol{k}}^{\mathsf{T}} \cdot \boldsymbol{M}_{\boldsymbol{k}}\} \exp\{-\frac{\beta}{2}(\hat{z} - \boldsymbol{M}_{\boldsymbol{k}})^{\mathsf{T}} \cdot (\hat{z} - \boldsymbol{M}_{\boldsymbol{k}})\} + C' \tag{22}$$

$$= -\log \sum_{k=1}^{N} \exp\{\beta\hat{z}^{\mathsf{T}} \cdot \boldsymbol{M}_{\boldsymbol{k}}\} \exp\{-\frac{\beta}{2}\hat{z}^{\mathsf{T}} \cdot \hat{z}\} + C' \tag{23}$$

$$= \frac{\beta}{2}\hat{z}^{\mathsf{T}} \cdot \hat{z} - \log \sum_{k=1}^{N} \exp\{\beta\hat{z}^{\mathsf{T}} \cdot \boldsymbol{M}_{\boldsymbol{k}}\} + C' \tag{24}$$

Up to an additive constant and a factor $\beta$, this expression is equivalent to the energy function proposed in [30]. According to the FEP formulation of PC, the neural dynamics performing iterative optimization of $\hat{z}$ can be derived from the gradient descent update with regard to the VFE. We start by deriving this gradient:

$$\nabla_{\hat{z}} F(\hat{z}) = \beta\hat{z} - \frac{\sum_{k=1}^{N} \exp\{\beta\hat{z}^{\mathsf{T}} \cdot \boldsymbol{M}_{\boldsymbol{k}}\} \cdot (\beta\boldsymbol{M}_{\boldsymbol{k}})}{\sum_{k'=1}^{N} \exp\{\beta\hat{z}^{\mathsf{T}} \cdot \boldsymbol{M}_{\boldsymbol{k'}}\}} \tag{25}$$

$$= \beta\{\hat{z} - \text{softmax}(\beta\hat{z}^{\mathsf{T}} \cdot \boldsymbol{M})\boldsymbol{M}^{\mathsf{T}}\} \tag{26}$$

Which yields the following update rule for $\hat{z}$:

$$\hat{z} \leftarrow \hat{z} + \alpha\beta\{\text{softmax}(\beta\hat{z}^{\mathsf{T}} \cdot \boldsymbol{M})\boldsymbol{M}^{\mathsf{T}} - \hat{z}\} \tag{27}$$

where $\alpha$ is the rate of the gradient descent. In particular, when $\alpha = \frac{1}{\beta}$ we obtain the update rule of the MCHN:

$$\hat{z} \leftarrow \text{softmax}(\beta\hat{z}^{\mathsf{T}} \cdot \boldsymbol{M})\boldsymbol{M}^{\mathsf{T}} \tag{28}$$

Looking back at the prior distribution $p(\boldsymbol{z}; \boldsymbol{M})$, the GMM is biased towards representations of larger Euclidean norm. This means that stored patterns $\boldsymbol{M}_{\boldsymbol{k}}$ aligned with other patterns of larger norms cannot attract the dynamics of the MCHN. This is represented in figure 3 where we have displayed two energy landscapes for an AM containing four 2D patterns. The VFE computed with a balanced GMM model (left) comprises a local minimum for each pattern, which is not the case for the VFE computed with the MCHN prior distribution.

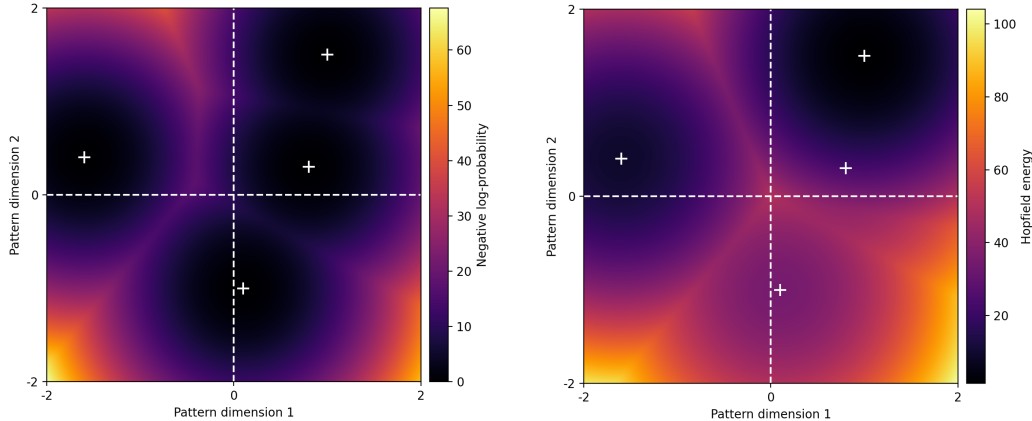

Figure 3: Illustration of the VFE landscape induced by the balanced (ours, left) and biased (MCHN, right) GMM prior distribution, for an AM of four 2D patterns represented in white. These figures were obtained with $\beta = 100$ and $\sigma = 0.2$.

## C   Derivations of the proposed models

### C.1   (VAE-)PC-GMM derivations

In this section, we provide derivations for the PC-GMM (and VAE-PC-GMM when $\hat{z}$ is initialized with the VAE encoder). We start by expressing the VFE using the GMM prior distribution:

$$
\begin{aligned}
F(\boldsymbol{x}, \hat{\boldsymbol{h}}_1, \ldots, \hat{\boldsymbol{h}_{L-1}}, \hat{\boldsymbol{z}}) = \sum_{l=0}^{L-1} \frac{1}{2} \|\hat{\boldsymbol{h}}_l - \boldsymbol{f}_{\boldsymbol{\theta}}^l(\hat{\boldsymbol{h}_{l+1}})\|^2 \\
- \log \sum_{k=1}^{N} \exp\{-\frac{1}{2\sigma^2}(\hat{\boldsymbol{z}} - \boldsymbol{M_k})^{\mathsf{T}} \cdot (\hat{\boldsymbol{z}} - \boldsymbol{M_k})\} \\
+ C'
\end{aligned}
\tag{29}
$$

where the constant C' contains other terms coming from the GMM expression that do not depend on $\{\boldsymbol{x}, \hat{\boldsymbol{h}}_1, \ldots, \hat{\boldsymbol{h}_{L-1}}, \hat{\boldsymbol{z}}\}$. According to the FEP formulation of PC, the neural dynamics simulate a gradient descent on this energy function. We can thus derive the update rules for the approximate posterior means $\{\hat{\boldsymbol{h}}_1, \ldots, \hat{\boldsymbol{h}_{L-1}}, \hat{\boldsymbol{z}}\}$. For all $1 \le l \le L$:

$$
\hat{\boldsymbol{h}}_l \leftarrow \hat{\boldsymbol{h}}_l - \alpha \nabla_{\hat{h}_l} F(\boldsymbol{x}, \hat{\boldsymbol{h}}_1, \ldots, \hat{\boldsymbol{h}_{L-1}}, \hat{\boldsymbol{z}})
\tag{30}
$$

where $\alpha$ is the rate of the gradient descent. For the intermediate layers $1 \le l < L$, we obtain the following update rule:

$$
\hat{\boldsymbol{h}}_l \leftarrow \hat{\boldsymbol{h}}_l - \underbrace{\alpha(\hat{\boldsymbol{h}}_l - \boldsymbol{f}_{\boldsymbol{\theta}}^l(\hat{\boldsymbol{h}_{l+1}}))}_{\text{Top-down}} + \underbrace{\alpha \boldsymbol{f}_{\boldsymbol{\theta}}^{l-1'}(\hat{\boldsymbol{h}}_l) \cdot (\hat{\boldsymbol{h}_{l-1}} - \boldsymbol{f}_{\boldsymbol{\theta}}^{l-1}(\hat{\boldsymbol{h}}_l))}_{\text{Bottom-up}}
\tag{31}
$$

This update rule combines top-down information pulling $\hat{\boldsymbol{h}}_l$ towards its prediction coming from the upper layer, and bottom-up information pulling it towards a value that reduces the prediction error on the lower layer. It is useful to introduce the notation $\epsilon_l = \hat{\boldsymbol{h}}_l - \boldsymbol{f}_{\boldsymbol{\theta}}^l(\hat{\boldsymbol{h}_{l+1}})$ called the prediction error on layer $l$. In the PC theory, at each layer a population of neurons encodes this quantity, while another encodes the current estimate $\hat{\boldsymbol{h}}_l$. For the last layer, the bottom-up signal is identical, but the top-down signal pulls $\hat{\boldsymbol{z}}$ towards values that maximize the prior $p(\boldsymbol{z})$:

$$\hat{\boldsymbol{z}} \leftarrow \hat{\boldsymbol{z}} + \underbrace{\frac{\alpha}{\sigma^2}\Big(\mathrm{softmax}\big(-\frac{\|\hat{\boldsymbol{z}}-\boldsymbol{M}\|_2^2}{2\sigma^2}\big)\cdot\boldsymbol{M}^{\mathsf{T}} - \hat{\boldsymbol{z}}\Big)}_{\text{Top-down}} + \underbrace{\alpha\big(\boldsymbol{f}_{\boldsymbol{\theta}}^{L-1\,'}(\hat{\boldsymbol{z}})\cdot\boldsymbol{\epsilon}_{L-1}\big)}_{\text{Bottom-up}} \qquad (32)$$

These update rules can be applied iteratively, which results in a dynamical system viewed in the PC theory as an RNN. Figure 4 represents this RNN unfolded in time (right) along with the assumed hierarchical probabilistic model (left). Algorithm 1 describes the forward pass through this PC network.

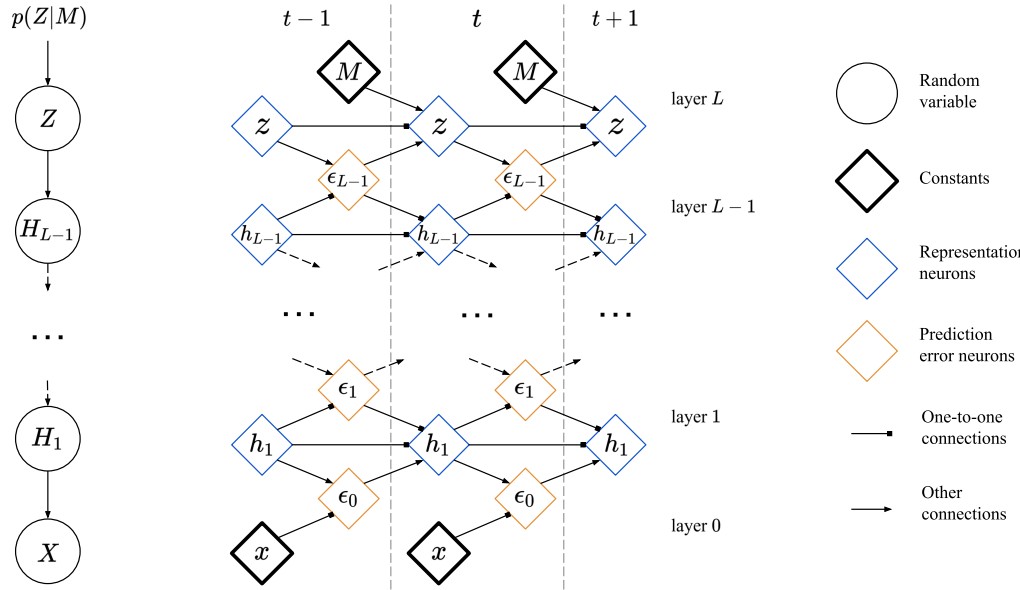

Figure 4: Left: probabilistic graphical model of our system. Right: corresponding PC network.

---

**Algorithm 1:** PC-GMM Memory retrieval

---

**Parameters:** $\boldsymbol{\theta}, \alpha, \sigma, \boldsymbol{M}$
**Input:** $\boldsymbol{x}$
Initialize $\big\{\hat{\boldsymbol{z}}, \hat{\boldsymbol{h}}_{L-1}, \ldots, \hat{\boldsymbol{h}}_1\big\}$
**for** $0 \le t < T$ **do**
    /* Compute prediction errors                                  */
    **for** $0 \le l < L$ **do**
        $\boldsymbol{\epsilon}_l \leftarrow \hat{\boldsymbol{h}}_l - \boldsymbol{f}_{\boldsymbol{\theta}}^l(\hat{\boldsymbol{h}}_{l+1})$
    **end**
    /* Update hidden representations                     */
    **for** $1 \le l < L$ **do**
        $\hat{\boldsymbol{h}}_l \leftarrow \hat{\boldsymbol{h}}_l + \alpha\big(\boldsymbol{f}_{\boldsymbol{\theta}}^{l-1\,'}(\hat{\boldsymbol{h}}_l)\cdot\boldsymbol{\epsilon}_{l-1} - \boldsymbol{\epsilon}_l\big)$
    **end**
    /* Update z                                                 */
    $\hat{\boldsymbol{z}} \leftarrow \hat{\boldsymbol{z}} + \alpha\big(\boldsymbol{f}_{\boldsymbol{\theta}}^{L-1\,'}(\hat{\boldsymbol{z}})\cdot\boldsymbol{\epsilon}_{L-1}\big) + \frac{\alpha}{\sigma^2}\Big(\mathrm{softmax}\big(-\frac{\|\hat{\boldsymbol{z}}-\boldsymbol{M}\|_2^2}{2\sigma^2}\big)\cdot\boldsymbol{M}^{\mathsf{T}} - \hat{\boldsymbol{z}}\Big)$
**end**

---

### C.2 (VAE-)BP-GMM derivations

Here we show that the PC-GMM dynamics approximate the gradient descent updates resulting from the application of BP on the loss function of the BP-GMM model. To obtain this result, we must

assume that the predictions $f_{\theta}^l(\hat{h_{l+1}})$ remain constant during this iterative inference process. This hypothesis, often called "fixed prediction assumption", is required to prove this results. Note that in our PC-GMM algorithm, we have updated the predictions at each iteration, so the result proven here do not apply in our case. Still this result is interesting as it highlights the relationship between PC-based inference and BP-based inference. Therefore, we assume here that the predictions $f_{\theta}^{l-1}(\hat{h}_l)$ and the derivatives $f_{\theta}^{l-1'}(\hat{h}_l)$ are fixed during the inference process. Only the prediction errors $\epsilon_l$ and the approximate posterior means $\hat{h}_l$ are updated. We recall that the loss function used in the BP-GMM model is defined as:

$$\mathcal{L}(\boldsymbol{x}, \hat{\boldsymbol{z}}) = \|\boldsymbol{f_\theta}(\hat{\boldsymbol{z}}) - \boldsymbol{x}\|^2 - \gamma \log p(\hat{\boldsymbol{z}}; \boldsymbol{M}) \tag{33}$$

Given an input $\boldsymbol{x}$ and a memory matrix $\boldsymbol{M}$, the dynamics of the PC network will reach equilibrium when for all $l$, $\nabla_{\hat{h}_l} F = 0$. For the intermediate layers, this is verified when:

$$\epsilon_l = \boldsymbol{f_\theta}^{l-1'}(\hat{h}_l) \cdot \epsilon_{l-1} \tag{34}$$

Equivalently, we can derive the expression of the gradients provided by BP on the intermediate quantities $\hat{h}_l$. On the bottom layer, we have:

$$\nabla_{\hat{h}_1} \|\boldsymbol{f_\theta}(\hat{\boldsymbol{z}}) - \boldsymbol{x}\|_2^2 = \nabla_{\hat{h}_1} \|\boldsymbol{f_\theta^0}(\hat{h}_1) - \boldsymbol{x}\|_2^2 \tag{35}$$
$$= 2\boldsymbol{f_\theta^{0'}}(\hat{h}_1) \cdot (\boldsymbol{f_\theta^0}(\hat{h}_1) - \boldsymbol{x}) \tag{36}$$
$$= -2\boldsymbol{f_\theta^{0'}}(\hat{h}_1) \cdot \epsilon_0 \tag{37}$$
$$= -2\epsilon_1 \tag{38}$$

Using the chain rule, we can derive the gradient with regard to $\hat{h}_l$ based on the gradient with regard to $\hat{h_{l-1}}$. We observe that we obtain the same recurrence relation between gradients $\nabla_{\hat{h}_l} \|\boldsymbol{f_\theta}(\hat{\boldsymbol{z}}) - \boldsymbol{x}\|_2^2$ than the one we obtained with prediction errors $\epsilon_l$ at equilibrium (equation 34):

$$\nabla_{\hat{h}_l} \|\boldsymbol{f_\theta}(\hat{\boldsymbol{z}}) - \boldsymbol{x}\|_2^2 = \boldsymbol{f_\theta}^{l-1'}(\hat{h}_l) \cdot \nabla_{\hat{h_{l-1}}} \|\boldsymbol{f_\theta}(\hat{\boldsymbol{z}}) - \boldsymbol{x}\|_2^2 \tag{39}$$

Therefore, according to the induction principle, we can conclude that for all layers $1 \le l < L$:

$$\epsilon_l = -2\nabla_{\hat{h}_l} \|\boldsymbol{f_\theta}(\hat{\boldsymbol{z}}) - \boldsymbol{x}\|_2^2 \tag{40}$$

Now, looking at the topmost layer, we can compare the update rule for $\hat{\boldsymbol{z}}$ prescribed by BP and PC. For PC, we have seen that the update rule is:

$$\hat{\boldsymbol{z}} \leftarrow \hat{\boldsymbol{z}} + \underbrace{\frac{\alpha}{\sigma^2}\left(\text{softmax}\left(-\frac{\|\hat{\boldsymbol{z}} - \boldsymbol{M}\|_2^2}{2\sigma^2}\right) \cdot \boldsymbol{M}^\intercal - \hat{\boldsymbol{z}}\right)}_{\text{Top-down}} + \underbrace{\alpha\left(\boldsymbol{f_\theta}^{L-1'}(\hat{\boldsymbol{z}}) \cdot \epsilon_{L-1}\right)}_{\text{Bottom-up}} \tag{41}$$

For BP, we once again use the chain rule:

$$\nabla_{\hat{\boldsymbol{z}}} \mathcal{L} = \nabla_{\hat{\boldsymbol{z}}}\left(-\gamma \log p(\hat{\boldsymbol{z}}; \boldsymbol{M})\right) + \boldsymbol{f_\theta}^{L-1'}(\hat{\boldsymbol{z}}) \cdot \nabla_{\hat{h_{L-1}}} \|\boldsymbol{f_\theta}(\hat{\boldsymbol{z}}) - \boldsymbol{x}\|_2^2 \tag{42}$$
$$= \underbrace{-\frac{\gamma}{\sigma^2}\left(\text{softmax}\left(-\frac{\|\hat{\boldsymbol{z}} - \boldsymbol{M}\|_2^2}{2\sigma^2}\right) \cdot \boldsymbol{M}^\intercal - \hat{\boldsymbol{z}}\right)}_{\text{Top-down}} - \underbrace{2\boldsymbol{f_\theta}^{L-1'}(\hat{\boldsymbol{z}})\epsilon_{L-1}}_{\text{Bottom-up}} \tag{43}$$

Taking $\gamma = \alpha = 2$ yields the exact same iterative inference update rule for both approaches. If we remove the "fixed prediction assumption" this equivalence no longer stands. However, this proves that the two approaches are closely related. In practice, we found that the two models performed

similarly but that the PC-GMM approach was prohibitively slow to propagate information for very deep generative models.

The iterative algorithm is described in algorithm 2.

---

**Algorithm 2:** BP-GMM Memory retrieval

---

**Parameters:** $\boldsymbol{\theta}, \alpha, \sigma, \gamma, \boldsymbol{M}$
**Input:** $\boldsymbol{x}$
Initialize $\hat{\boldsymbol{z}}$
**for** $0 \leq t < T$ **do**

    /* Compute the prediction                                                 */
    $\hat{\boldsymbol{x}} \leftarrow \boldsymbol{f_\theta}(\hat{\boldsymbol{z}})$
    /* Compute the energy function                                     */
    $\mathcal{L} \leftarrow \|\hat{\boldsymbol{x}} - \boldsymbol{x}\|^2 - \gamma \log p(\hat{\boldsymbol{z}}; \boldsymbol{M})$
    /* Update z using BP                                                  */
    $\hat{\boldsymbol{z}} \leftarrow \hat{\boldsymbol{z}} - \alpha \nabla_{\hat{\boldsymbol{z}}} \mathcal{L}$

**end**

---

### C.3 VAE-GMM derivations

The derivation of the VAE-GMM model is straightforward. We simply remove the bottom-up information pathway of the VAE-PC-GMM model and instead consider that the amortized inference performed by the encoder already conveys the necessary information from $\boldsymbol{x}$. The update rule for $\hat{\boldsymbol{z}}$ becomes:

$$\hat{\boldsymbol{z}} \leftarrow \hat{\boldsymbol{z}} + \frac{\alpha}{\sigma^2} \left( \text{softmax}\left( -\frac{\|\hat{\boldsymbol{z}} - \boldsymbol{M}\|_2^2}{2\sigma^2} \right) \cdot \boldsymbol{M}^\intercal - \hat{\boldsymbol{z}} \right) \tag{44}$$

If we choose $\alpha = \sigma^2$, we obtain the update rule of the VAE-GMM model:

$$\hat{\boldsymbol{z}} \leftarrow \text{softmax}\left( -\frac{\|\hat{\boldsymbol{z}} - \boldsymbol{M}\|_2^2}{2\sigma^2} \right) \cdot \boldsymbol{M}^\intercal \tag{45}$$

## D Implementation details

All the presented experiments were performed on a single NVIDIA GeForce GTX 1060 GPU.

Training was performed on the training sets of the two datasets, and the results reported in this article were obtained on the testing sets.

The training hyperparameters (learning rate, number of steps for the MemN2N model) were optimized in order to achieve the lowest prediction error on the training set. The memory retrieval hyperparameters ($\sigma$ for GMM models, $\beta$ for MCHN models, $\gamma$ for the BP-GMM model) were optimized in order to achieve the highest successful retrieval percentage on the training set.

The reported results were obtained using one seed for the VAE, 5 seeds for the AM models that require training (MemN2N, NTM, VAE-GMM*), and 10 seeds for the memory retrieval scenarios that include randomness (noise and mask).

We provide the code including the implementation of the proposed models, our implementation of the benchmark models, the different memory retrieval scenarios and the hyperparameter values we have experimented with: `https://github.com/sino7/predictive_coding_associative_memories`.

Our implementation of the MONet model was adapted from the implementation provided in the github repository `https://github.com/baudm/MONet-pytorch`, and we used the provided pretrained weights on the CLEVR dataset.

# E  Additional results

## E.1  Ablation study

In this section, we investigate the impact of two features of our model: the initialization of $\hat{z}$ using the encoder, and the use of a balanced GMM instead of the biased GMM of the MCHN model (see appendix B).

We have reproduced the memory retrieval experiment with noisy inputs on two new model variations: BP-GMM (without VAE initialization) and VAE-BP-Hopfield. The VAE-BP-Hopfield is the biased GMM version of the VAE-BP-GMM model, where the BP is used to perform iterative inference in order to minimize the loss function:

$$\mathcal{L} = \|f_{\theta}(\hat{z}) - x\|^2 + \gamma\left(\frac{1}{2}\hat{z}^{\mathsf{T}} \cdot \hat{z} - \frac{1}{\beta}\log\sum_{k=1}^{N}\exp\{\beta\hat{z}^{\mathsf{T}} \cdot M_k\}\right) \tag{46}$$

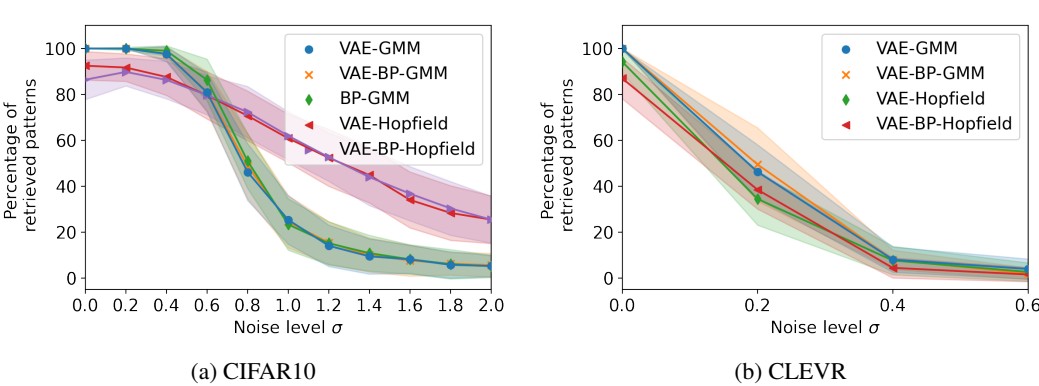

(a) CIFAR10              (b) CLEVR

Figure 5: Percentage of successful memory retrieval using inputs corrupted with a noise of varying standard deviation $\sigma$. Intervals indicate standard deviation.

We report in figure 5 the percentage of successful memory retrieval using these different models. On the CLEVR dataset, the BP-GMM (without VAE initialization) always failed to retrieve the correct memory pattern. On the CIFAR10 dataset, it performed exactly the same as the VAE-GMM and VAE-BP-GMM models. This argues in favor of the simpler VAE-GMM model, that seems to convey information from the input $x$ properly enough. This is also observed by comparing the results using the VAE-BP-Hopfield model and the simpler VAE-Hopfield model.

We can note that on the CIFAR10 dataset, Hopfield based retrieval is more robust to very high levels of noise. We believe that in this case, the bias towards patterns of high L2 norm might partially counter the indirect effect of the noise onto $\hat{z}$. On the CLEVR dataset this synergy is not observed and all models rapidly fail to retrieve patterns in memory.

On the other hand, we can see that Hopfield-based models never reach perfect retrieval even when the presented inputs are clean, which argues in favor of the balanced GMM alternative.

## E.2  Analysis of the trained VAE-GMM* precision coefficients

In this section, we investigate the effect of precision coefficient learning in the VAE-GMM* model. After training in two different scenarios: RGB rotation and shift, we compare the learned precision coefficients. For a more straightforward analysis, we have restricted the precision matrix to be diagonal. This way we can directly identify dimensions of the representation $z$ that are deemed more or less relevant for memory retrieval in both scenarios.

The CLEVR representation is structured into four object representations (one for the background and the three others for possible objects in the scene). We have identified four dimensions of the object representation where the precision coefficients in both scenarios presented the highest disagreement. We have then sampled an image from the CLEVR dataset and made variations along these dimensions

to observe their effect on the decoded images. As shown in figure 6, these four dimensions can be interpreted as encoding color, size and position.

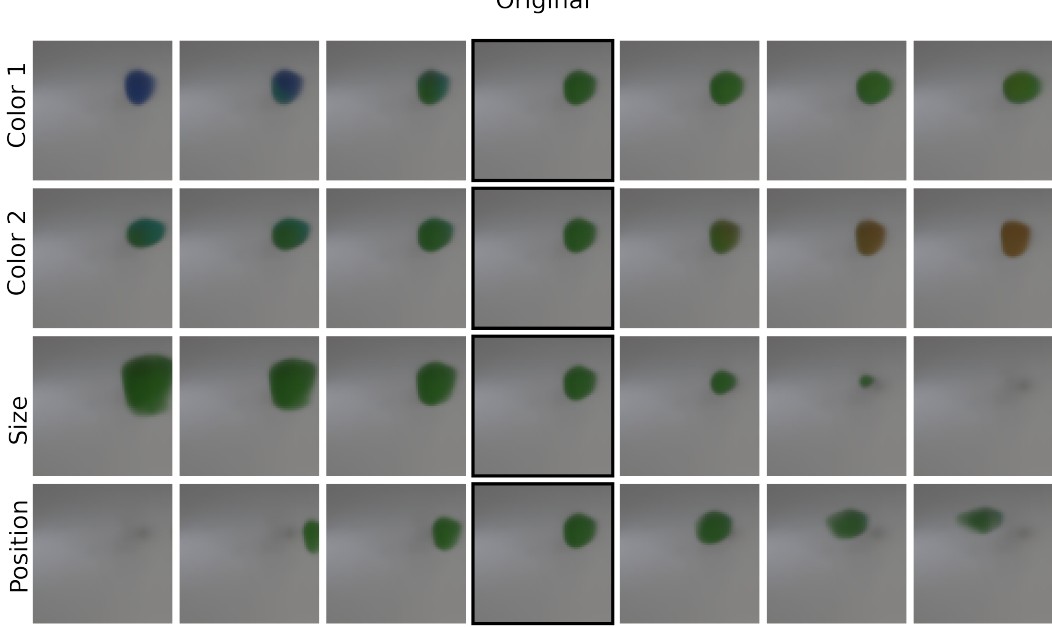

Figure 6: Varying the representation along some dimensions. The dimensions for the first two rows seem to encode object color. The dimension for row 3 seems to encode object size. The dimension for row 4 seems to encode object horizontal position.

For the VAE-GMM* model trained with the RGB rotation, the precision coefficients corresponding to the color dimension were lower, meaning that the retrieval mechanism gave less importance to these features when comparing the inferred representation $\hat{z}$ with the memory patterns $M_k$. Conversely, for the VAE-GMM* model trained with shifted images, the precision coefficients corresponding to the position and size (to a lower a extent) were lower. Consequently, the two trained models (with the same memory content) can react differently to the same input pattern, as shown in figure 7.

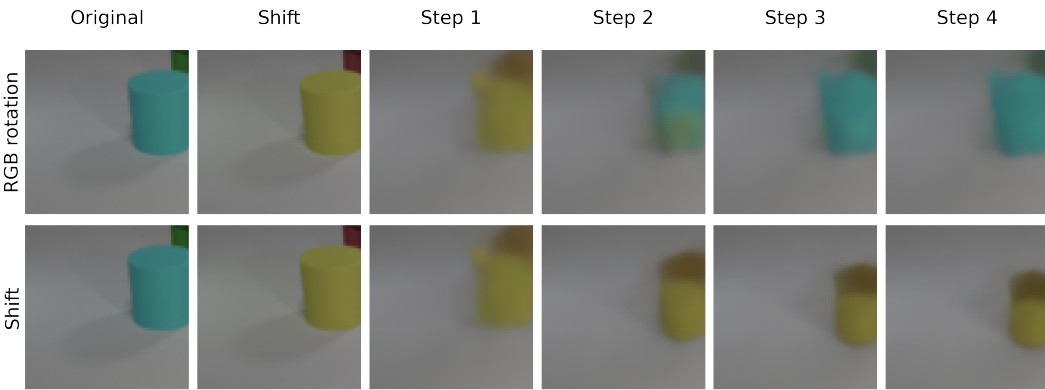

Figure 7: Images decoded from the representations $\hat{z}$ at different steps of the inference process, using the VAE-GMM* models trained with the RGB rotation (first row) and with shifted images (second row). For this experiment, we have use the VAE-GMM model with a lower update coefficient $\alpha$ (see equation 44) to observe a smooth convergence.

We can observe that for an input image obtained by applying an RGB rotation on one of the stored patterns, the model trained on the correct task properly retrieves the pattern, while the model trained with shifted images instead converges to a stored pattern corresponding to a similar object in a

different position. These results confirm our intuition that adaptation of the precision coefficients can help the proposed VAE-GMM model to give more or less importance to certain representation features for memory retrieval.

### E.3 One-shot generation

In this section, we display examples of images sampled with our memory-dependent generative model on the CLEVR dataset. The first row of figure 8 contains the input patterns written in the memory. The writing operation simply consists in encoding the images and building the memory matrix with the obtained column vectors. In the bottom of this figure are images sampled from the memory-dependent generative model. We can observe that the sampled images contain similar objects in the same positions, with slight variations of shape, size, position or color.

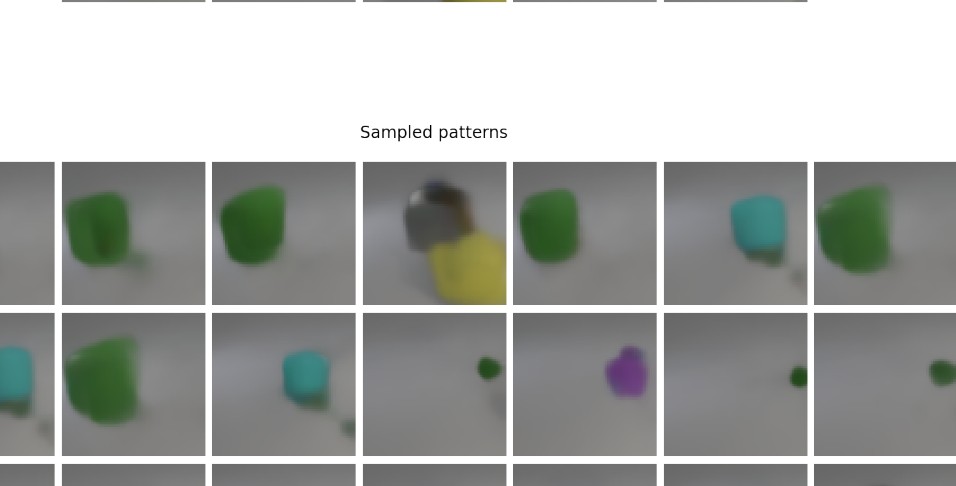

Figure 8: Examples of generated images.

### E.4 Capacity

We have tried measuring the retrieval success rate with a varying number of patterns in the memory. The models based on a balanced GMM can achieve 100% of successful retrievals when no noise is applied, for a better comparison, we thus experiment with input patterns corrupted with a noise of standard deviation $\sigma = 0.6$. We compare the performance of the GMM models with or without the representation component (respectively VAE-GMM and GMM) as well as the performance with the MCHN variants (VAE-Hopfield). Since the MCHN applied on the raw pixel level scores very low in our initial experiments (see table 2 and table 3), we only experiment here with the variant working on the representation level. This experiment is conducted on the CIFAR10 dataset, with memory stores of size varying from $N = 5$ to $N = 10000$.

The results are displayed in figure 9. We can observe that the performance of the VAE-Hopfield model drops faster than the performance of the models based on the balanced GMM implementation we proposed. Another result is that the use of a representation component does not seem to improve the capacity of the model.

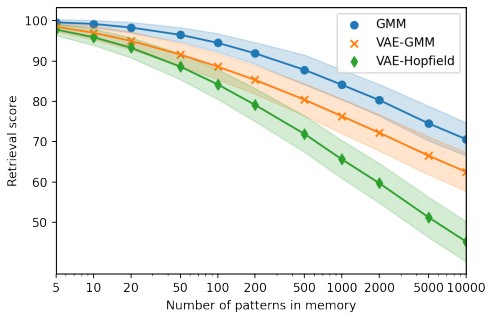

Figure 9: Evolution of the retrieval rate according to the number of memory patterns.

## E.5  Additional figures

In this section, we provide examples of input and retrieved images. On the CIFAR10 dataset, we provide examples using AM models working on the pixel level in figure 10 and AM models working on the representation level in figure 11. On the CLEVR dataset, we provide examples using models working on the representation level, with and without dedicated training in the "RGB rotation" scenario, in figure 12.

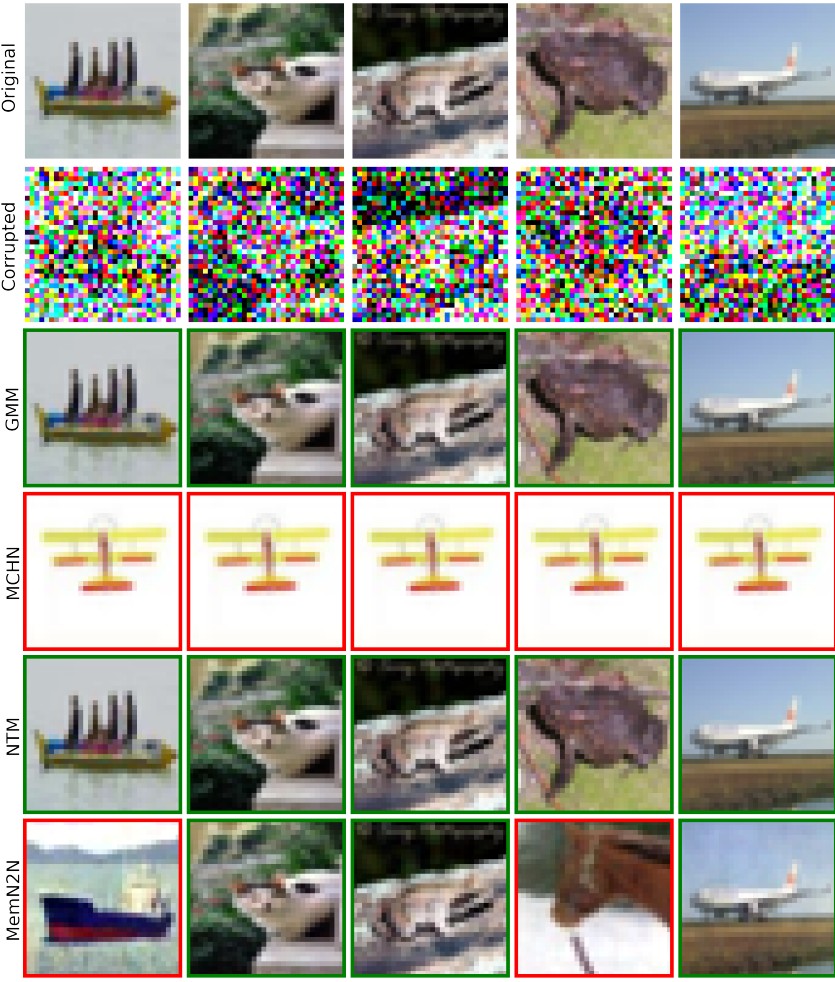

Figure 10: Retrieved images using inputs corrupted with a noise of standard deviation $\sigma = 0.6$.

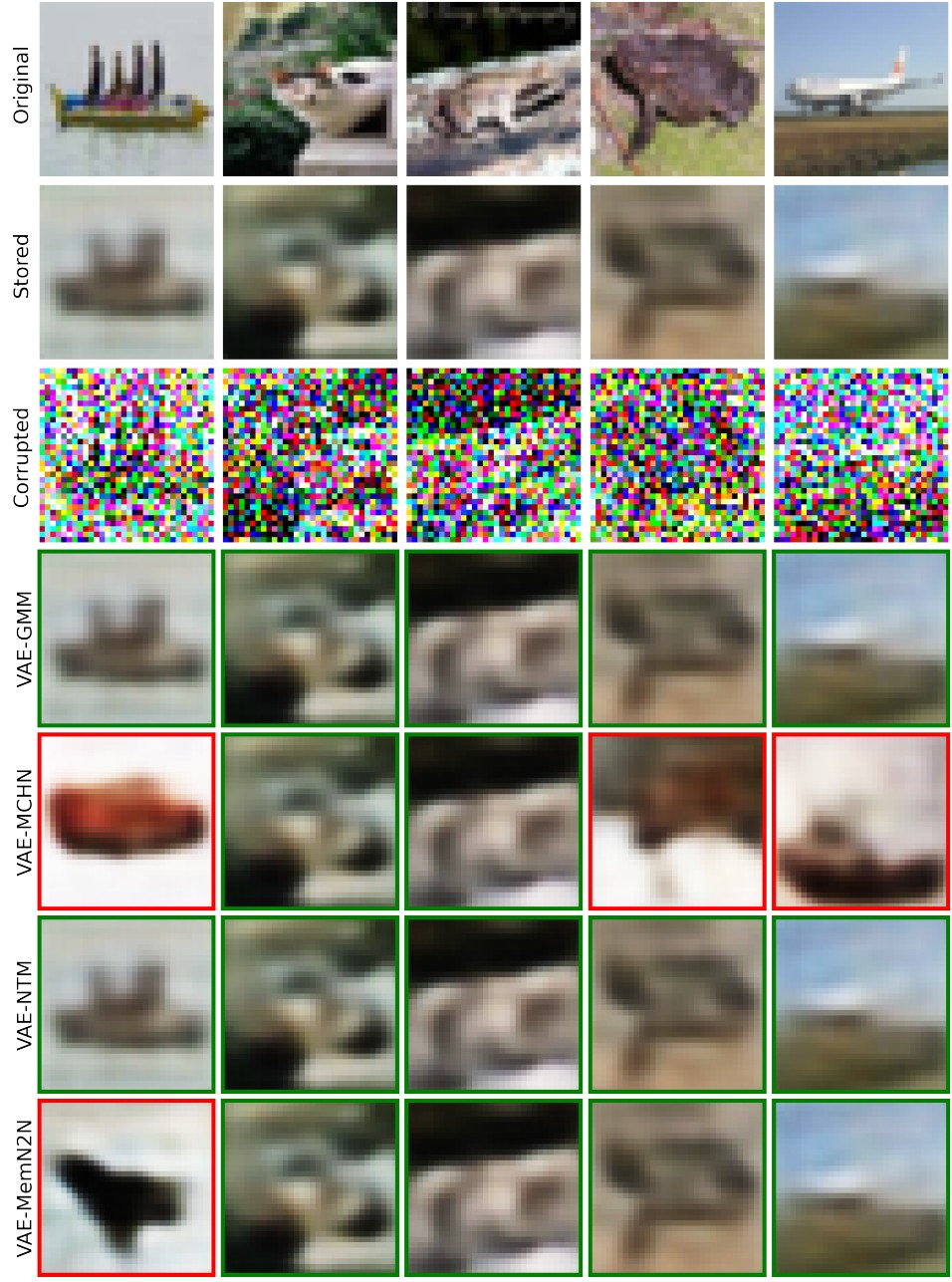

Figure 11: Examples of retrieved images with different models, using inputs corrupted with a noise of standard deviation $\sigma = 0.6$.

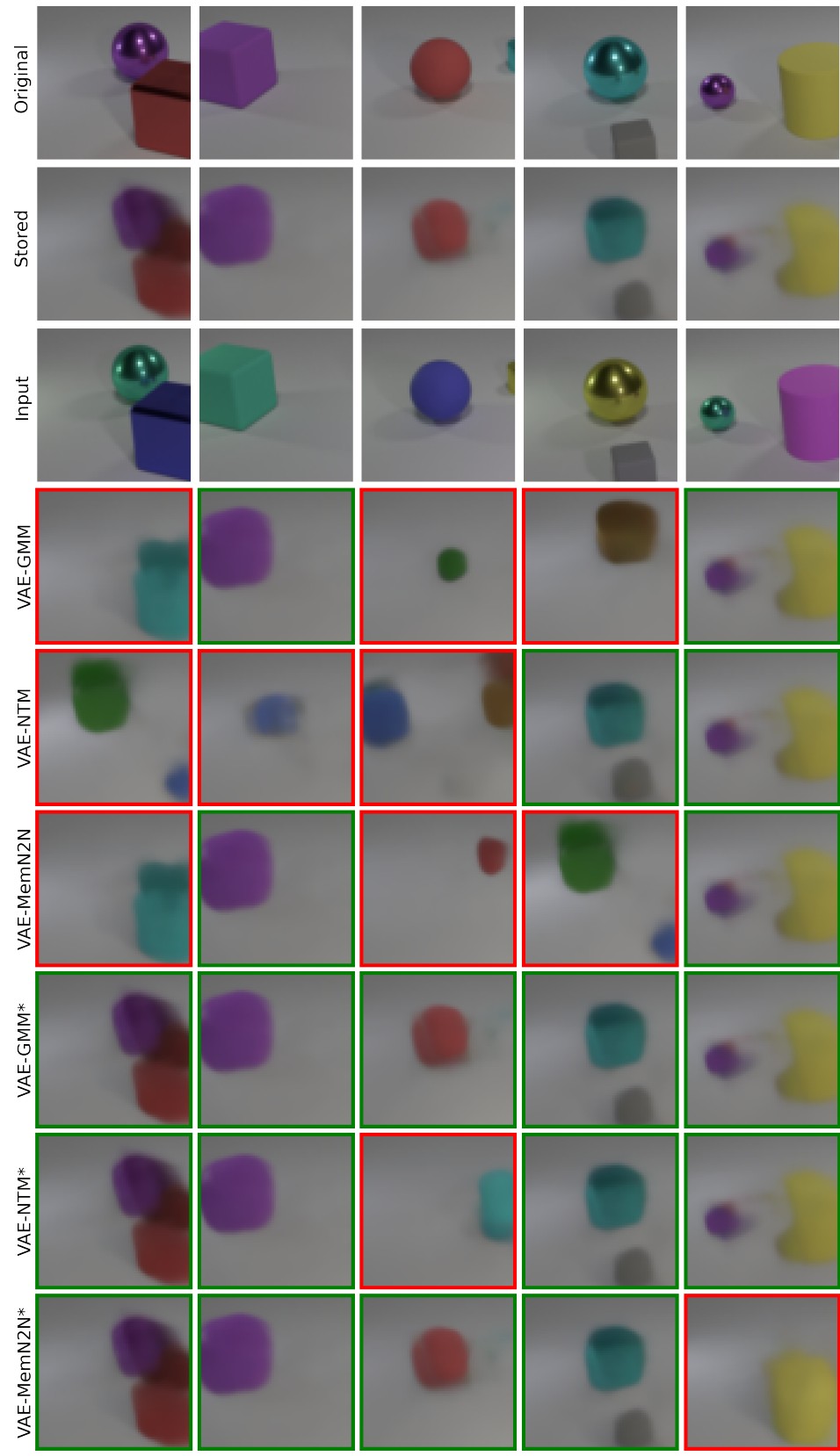

Figure 12: Retrieved images with different models in the RGB rotation scenario.