# OpenReview forum: "On the relationship between variational inference and auto-associative memory"
_NeurIPS.cc/2022/Conference — NeurIPS 2022 Accept_

### Official Review · Reviewer_NJ3P · 2022-06-20

**Rating:** 5
**Confidence:** 4
**Soundness:** 3 good
**Presentation:** 2 fair
**Contribution:** 3 good

**Summary:**

This work provides a new formalization of associative memory models, merging them with the framework of variational inference. I found the idea interesting: In the standard predictive coding (PC) literature, we aim to infer the 'cause' that generates a specific datapoint. This cause may, or may not, be also influenced by a prior. When modelling the prior distribution according to a dataset of memories M, converging to the 'cause' that minimizes the variational free energy can be seen as a memory retrieval process. In the specific case of the prior having a Gaussian mixture distribution, and when no hierarchy is present, the resulting model is basically equivalent to a modern Hopfield network. Interestingly, in section 4.4 the authors simplify the update rule of PC and use t to derive an update rule that is similar to that of universal Hopfield networks with Euclidean distance as a similarity function [1]. This introductory intuition paves a foundation that can be used to build over associative memory models, able to encode semantic information by using variational auto encoders (VAEs).  The experiments show that (1) scoring memories in a 'semantic' space improves most of the performance of standard memory models, and that (2) it is possible to train the covariance matrix underlying the GMM to further improve the results under two kinds of corruption.


[1] Millidge, Beren, et al. "Universal Hopfield Networks: A General Framework for Single-Shot Associative Memory Models." arXiv preprint arXiv:2202.04557 (2022).

**Questions:**

Regarding clarity, could the authors:

1) Provide sketches of the proposed models; I believe they would be a nice and needed improvement to the paper. Particularly, could you describe step by step the retrieval process of the VAE-BP-GMM model?

2) Why have you decided to put a lot of emphasis on PC in the preliminaries and introduction, and then remove it from any experiment? This separation gave me the feeling of reading two different works in the same paper: the first, provides a theoretical derivation of AM models based on the theory of variational inference and proposes a new hierarchical model based on PC and VAEs; the second performs multiple experiments without any PC in it. I do get the underlying connection between the two parts.

Random question that would make an interesting addition to the theoretical side of the paper, but not important in the scoring:

One of the interesting properties of predictive coding, is that it has a neural implementation, where all the operations are implemented using local information via inhibitory and excitatory connections. Is there a neural implementation for the Hopfield network variation of PC?






**Limitations:**

The limitations are correctly addressed in the last page of the paper.

**Strengths And Weaknesses:**

Strengths:

- The paper is sound, and I like the overall intuition;
- The experimental results improve the performance over the (well picked) baselines.

Cons:

- The writing becomes confusing in the methods section. I have had some troubles understanding the exact functioning and structure of the proposed models. It would be nice to have drawings that explain how these models retrieve stored memories, as this confusion made me re-read sections 4 and 5 several times.

- The authors show multiple experiments that compare standard models that perform storage and retrieval operations directly on the pixel space, against the proposed ones.  I'm puzzled by the choice of the authors on this task: no one of the proposed models is better than the baseline (Tabs. 2 and 3).

- The authors then conclude with an interesting result of the paper: training on the retrieval task improves the performance. Again, however, the proposed method outperforms the competitor only on one task.

- An experimental contribution, as stated by the authors, is that distance-based similarity scores seem to always outperform dot products. This is has been already shown in [1] cited above: Manhattan distance seems even better than Euclidean distance on natural images.

All in all, I believe this is a nice paper, where the strengths slightly outperform the weaknesses, and that will be eventually accepted in a high level conference with little effort by addressing mine and other reviewers comments. While I believe that with little work this can be made a stronger submission, this paper in my opinion falls slightly above the acceptance rate, and hence I propose weak accept.

---

> ### Author Response · Authors · 2022-08-02
> **Response to Reviewer NJ3P**
>
> Thank you for the time spent on our submission and for your comments.
>
> We answer here the points you have raised:
>
> - The authors show multiple experiments that compare standard models that perform storage and retrieval operations directly on the pixel space, against the proposed ones. I'm puzzled by the choice of the authors on this task: no one of the proposed models is better than the baseline (Tabs. 2 and 3).
>
> The proposed AM model labeled GMM performs best in multiple scenarios, while the NTM/VAE-NTM performs best in others. The contribution of this article is mainly theoretical, we do not claim that the proposed models outperform other benchmarks, the experiments only validate that the models built with the proposed framework can perform memory retrieval relatively well in comparison with existing models.
>
> Also, we can argue that the proposed models have several advantages compared to NTMs: they have a theoretical foundation, and the retrieval processes do not need to be trained (except for the GMM* version).
>
> - An experimental contribution, as stated by the authors, is that distance-based similarity scores seem to always outperform dot products. This is has been already shown in [1] cited above: Manhattan distance seems even better than Euclidean distance on natural images.
>
> This is right, we cited this article and indeed our results corroborate theirs. Note however that they implement distance based similarity as the inverse of distance, while we use the negative distance (see line 174-180 of their linked code https://github.com/BerenMillidge/Theory_Associative_Memory/blob/main/functions.py). It would be interesting in future work to evaluate the Manhattan distance variant of our models.
>
> - Provide sketches of the proposed models; I believe they would be a nice and needed improvement to the paper. Particularly, could you describe step by step the retrieval process of the VAE-BP-GMM model?
>
> We have added a figure representing the reading mechanisms of the different proposed models. We have also added an algorithm in the Supplementary Materials to clarify the retrieval process of the VAE-BP-GMM. We hope that this helps improving the clarity of our paper, and welcome any further suggestion.
>
> - Why have you decided to put a lot of emphasis on PC in the preliminaries and introduction, and then remove it from any experiment? This separation gave me the feeling of reading two different works in the same paper: the first, provides a theoretical derivation of AM models based on the theory of variational inference and proposes a new hierarchical model based on PC and VAEs; the second performs multiple experiments without any PC in it. I do get the underlying connection between the two parts.
>
> We first approached this subject from the FEP/PC perspective so it felt natural to put some emphasis on it. The ideas of adapting precision matrices and using BP for inference both come from the PC literature. Although we did not present experimental results with the VAE-PC-GMM model, it still serves as an intermediary step for the design of the VAE-BP-GMM model in the train of thought we try to convey in the article. Still, we moved the figure representing the VAE-PC-GMM model to the Supplementary Materials to put less emphasis on this particular model.
>
> - One of the interesting properties of predictive coding, is that it has a neural implementation, where all the operations are implemented using local information via inhibitory and excitatory connections. Is there a neural implementation for the Hopfield network variation of PC?
>
> This is an interesting question. We have voluntarily avoided mentioning this as a potential strength of the PC-based model because the GMM-based update we have derived (last equation of alg. 1) is not standard in the PC literature and we have not tried to look for potential biologically plausible implementations.

---

> > ### Comment · Reviewer_NJ3P · 2022-08-08
> > **Acknowledge of the rebuttals**
> >
> > Thank you for your answer and clarification. I am still undecided about by final decision, and hence I won't increase my score (still, I suggest weak accept).

---

> > > ### Author Response · Authors · 2022-08-08
> > > **Response to Reviewer NJ3P**
> > >
> > > If by any chance this was a mistake from your part, please note that the score of 5 you gave us corresponds to the mention "borderline accept". If you meant to recommend "weak accept" as said in your comments, this should correspond to a score of 6.

---

### Official Review · Reviewer_7s9W · 2022-07-11

**Rating:** 5
**Confidence:** 4
**Soundness:** 2 fair
**Presentation:** 3 good
**Contribution:** 2 fair

**Summary:**

This paper integrates the Modern Continuous Hopfield Network into the latent space of a VAE and shows the relationships between this, predictive coding, and the free energy principle. It then provides provisional analysis of these different approaches on two baseline datasets.

**Questions:**

See above.

**Limitations:**

Yes.

**Strengths And Weaknesses:**

Should cite https://pubmed.ncbi.nlm.nih.gov/33067397/ and have as a benchmark this model that is just a normal VAE. This may also make your classification in Table 1 harder for saying the VAE is not associative.

Is the relationship to FEP really important here? Why not cache everything in terms of ELBO and VAEs that more people should be familiar with?

There is no no intuitive justification for biasing the GMM prior towards patterns with a larger Euclidian norm?

Why was MONet used for the CLEVR dataset if it makes vector similarity metrics less meaningful. You say that you try to address this by limiting the number of items etc. but why not just use another pretrained VAE where this problem is removed entirely?

When you say representation you mean learning a compressed latent embedding as opposed to working with the raw image pixels? This could be made more clear.

You say that the VAE embeddings are not necessary for strong performance in Tables 2 and 3 but this is very unsurprising given that you only use n=100 images. Please add analysis where the performance of each model changes as a function of the number of memories being stored.

What is the difference between the VAE-Hopfield and the VAE-GMM? In using the softmax update rule for the GMM is this not the same as the MCHN update rule?

This paper should be another benchmark: http://proceedings.mlr.press/v80/marino18a/marino18a.pdf it also makes the point of a hierarchical model integrating top down and bottom up signals in a PC esque way.

This work should also be cited: https://arxiv.org/pdf/1709.07116.pdf

Here is associative memory being read from in a probabilistic fashion and iteratively: https://arxiv.org/pdf/1804.01756.pdf while this work https://proceedings.neurips.cc/paper/2021/file/8171ac2c5544a5cb54ac0f38bf477af4-Paper.pdf shows that SDM is closely related to Attention and thus MCHNs. Also see the Kanerva Machine follow up with more of an explicit discussion of convergence to attractor states here: https://proceedings.neurips.cc/paper/2018/file/6e4243f5511fd6ef0f03e9f386d54403-Paper.pdf

You claim that distance based metrics are better than taking dot products but this seems like a false dichotomy. If you normalize your vectors first then you can use a dot product and have cosine similarity etc. Modern Hopfield Networks use a form of bounded normalization (https://arxiv.org/pdf/1606.01164.pdf, pg.5), MCHNs also discuss normalization extensively and how it is intrinsic to the bipolar bits used in the original Hopfield Networks, Transformers use LayerNorm, it seems unfair to characterize MCHNs as not performing well only because you decided to not L2 norm their vectors. Also the Universal Hopfield Networks paper you cite shows that different distance metrics are heterogenous in their performance.

I'd also like for it to be more clear what the implications or benefits of this work are? In many cases it looks like the NTM does better and there is not much discussion of this while you also acknowledge that the NTM is more flexible with writing and reading mechanisms. If the objective is to be able to store as many things as possible then there should be analysis of memory capacity. If the goal is just to provide a novel insight well fixed points in memory and representing them as GMMs is not novel nor is relating to them to VAEs (see https://jamied157.github.io/HQA/) which cites work from Hinton in 1994 on this topic (see the other references I have also provided above).

I do like Eqs. 7 and 8. But these come from your axioms in 3.2 including setting the MCHN to use the softmax update from the start and so these convergences between the MCHN loss function and the VFE should not be surprising? This brings me back to the lack of an explanation for weighting each gaussian by the Euclidian norm of the memory. And my critique about not having a comparison where these vectors are normalized.

The discussion of VAE-PC-GMM also felt strange given that you conclude it is computationally intractable. And this is probably the gold standard model that you build towards in the paper.

To conclude, at a minimum I'd like to see experiments around memory capacity, an explanation for weighting of the Gaussians, a much larger related work section, and normalization of the memories. At a maximum, I'd like to know what makes this work unique in the context of both new insights not made in the context of the papers I reference and or new baseline beating performance on ML tasks.

---

> ### Author Response · Authors · 2022-08-02
> **Response to Reviewer 7s9W (1)**
>
> Thank you for the time spent on reviewing our work and for your comments.
>
> We answer here the different points you have raised:
>
> - Should cite https://pubmed.ncbi.nlm.nih.gov/33067397/ and have as a benchmark this model that is just a normal VAE. This may also make your classification in Table 1 harder for saying the VAE is not associative.
>
> We have added this model to the related work, as well as the literature regarding Kanerva Machines and GMM based VAEs, and updated table 1. The comparison with this model might be unfair, as training is necessary in order to store patterns as attractors. In comparison, our models allow us to directly write patterns as columns of the memory matrix.
>
> - Is the relationship to FEP really important here? Why not cache everything in terms of ELBO and VAEs that more people should be familiar with?
>
> The relationship with the FEP is important as it is the framework in which PC is usually formulated. The precision weighting variant presented in the last Methods subsection is also part of the FEP/PC toolbox. Additionally, in ML, the ELBO is usually optimized through learning alone, while in the FEP literature it is common to minimize VFE by optimizing $\hat{z}$. Finally, the fact that VFE is an energy function being minimized instead of the ELBO being maximized, while equivalent, makes the connection with MCHN energy function subjectively more natural.
>
> - There is no no intuitive justification for biasing the GMM prior towards patterns with a larger Euclidean norm?
>
> This is our point, this bias has no justification (see lines 161-164 of the original submission). This is the reason why we propose other models with balanced mixing coefficients (the models with the suffix GMM). This bias leads to a dot-product attention update rule, while the unbiased version leads to the Euclidean distance attention update rule used in our models.
>
> - Why was MONet used for the CLEVR dataset if it makes vector similarity metrics less meaningful. You say that you try to address this by limiting the number of items etc. but why not just use another pretrained VAE where this problem is removed entirely?
>
> Knowing that MONet encodes different objects separately provides easily interpretable embeddings with identifiable features. We chose this encoder especially for the last experiment where we show that learning precision coefficients allows to obtain a memory retrieval mechanism that can weight differently the different features of the embedding. If instead we had an entangled representation, not factored according to objects, we suspect that the retrieval rates of the GMM*, NTM* and MEMN2N* models in the last table would be way lower.
>
> - When you say representation you mean learning a compressed latent embedding as opposed to working with the raw image pixels? This could be made more clear.
>
> The representation is indeed the value of the latent variable $Z$ in our article.
>
> - You say that the VAE embeddings are not necessary for strong performance in Tables 2 and 3 but this is very unsurprising given that you only use n=100 images. Please add analysis where the performance of each model changes as a function of the number of memories being stored.
>
> We agree that this would be a valuable additional experiment, as stated in the discussion about the limitations of our work. Please correct us if we are wrong, but your comment implies that you believe that retrieval scores would deteriorate faster with raw pixel images compared with embeddings, when n increases. What makes you think that ? This prediction is not obvious to us. We are currently running this additional experiment, and we will incorporate the results in the Supplementary Materials.
>
> - What is the difference between the VAE-Hopfield and the VAE-GMM? In using the softmax update rule for the GMM is this not the same as the MCHN update rule?
>
> Again, this is explained in the lines 161-164 of the original submission. The proposed models use a balanced GMM while the VAE-Hopfield uses a biased GMM prior distribution. The balanced GMM translates into Euclidean distance attention update rules, while the biased GMM translates into the dot-product attention update rule of MCHNs.
>
> - This paper should be another benchmark: http://proceedings.mlr.press/v80/marino18a/marino18a.pdf it also makes the point of a hierarchical model integrating top down and bottom up signals in a PC esque way.
>
> This is an interesting read but not really a comparable work since it makes no mention of AM nor does it use a memory dependent generative model. Still, their "learning to optimize" approach could be interesting in order to improve the convergence speed of our iterative methods. We added this point in the discussion.
>
> - This work should also be cited: https://arxiv.org/pdf/1709.07116.pdf
>
> We have added this work as well as another article using GMM together with VAEs in the related work section.

---

> > ### Author Response · Authors · 2022-08-02
> > **Response to Reviewer 7s9W (2)**
> >
> > - Here is associative memory being read from in a probabilistic fashion and iteratively: https://arxiv.org/pdf/1804.01756.pdf while this work https://proceedings.neurips.cc/paper/2021/file/8171ac2c5544a5cb54ac0f38bf477af4-Paper.pdf shows that SDM is closely related to Attention and thus MCHNs. Also see the Kanerva Machine follow up with more of an explicit discussion of convergence to attractor states here: https://proceedings.neurips.cc/paper/2018/file/6e4243f5511fd6ef0f03e9f386d54403-Paper.pdf
> >
> > Indeed, all the literature around SDMs and Kanerva Machines is very relevant to our work. A paragraph has been added in the Related Work section, and a short paragraph added to the Discussion section.
> >
> > - You claim that distance based metrics are better than taking dot products but this seems like a false dichotomy. If you normalize your vectors first then you can use a dot product and have cosine similarity etc. Modern Hopfield Networks use a form of bounded normalization (https://arxiv.org/pdf/1606.01164.pdf, pg.5), MCHNs also discuss normalization extensively and how it is intrinsic to the bipolar bits used in the original Hopfield Networks, Transformers use LayerNorm, it seems unfair to characterize MCHNs as not performing well only because you decided to not L2 norm their vectors. Also the Universal Hopfield Networks paper you cite shows that different distance metrics are heterogenous in their performance.
> >
> > This is a fair point, cosine similarity should perform better than dot-product similarity, and this can be obtained with normalization. However, normalizing each pattern by its norm removes part of the information (aligned patterns are now equal), although this is arguably not significant. Regarding methods that apply the same linear normalization to all patterns, they do not remove the fact that after normalization, the AM will be biased towards normalized patterns of higher Euclidean norm.
> >
> > In comparison, the Euclidean distance attention does not require any additional trick in order to properly retrieve stored patterns, and is theoretically sound since we derived it from our memory-dependent generative model.
> >
> > We have nuanced this claim in the discussion.
> >
> > - I'd also like for it to be more clear what the implications or benefits of this work are? In many cases it looks like the NTM does better and there is not much discussion of this while you also acknowledge that the NTM is more flexible with writing and reading mechanisms. If the objective is to be able to store as many things as possible then there should be analysis of memory capacity. If the goal is just to provide a novel insight well fixed points in memory and representing them as GMMs is not novel nor is relating to them to VAEs (see https://jamied157.github.io/HQA/) which cites work from Hinton in 1994 on this topic (see the other references I have also provided above).
> >
> > The goal is indeed to provide novel insights from the application of PC methods to AM modeling. While there is nothing new about attractor dynamics for AMs, using GMMs or relating them to VAEs, this work is the first, to our knowledge, to approach the problem from the PC perspective. We believe that the PC toolbox can bring novel ideas to AM modeling, which can later inspire other advances in this field. Especially, the PC and BP based models we have proposed provide bottom-up information throughout the whole iterative memory retrieval process, which is something that other models are not considering. Additionally, the idea of optimizing the precision matrices (inverse covariance) of the generative model comes from the PC literature. Finally, relating MCHNs to PC is also a novel insight brought by this work.
> >
> > - I do like Eqs. 7 and 8. But these come from your axioms in 3.2 including setting the MCHN to use the softmax update from the start and so these convergences between the MCHN loss function and the VFE should not be surprising? This brings me back to the lack of an explanation for weighting each gaussian by the Euclidean norm of the memory. And my critique about not having a comparison where these vectors are normalized.
> >
> > There is indeed nothing surprising there. Our approach has been to study existing models and examine whether they would fit the proposed variational inference framework. We have noticed the the MCHN could fit our framework and found that the corresponding prior distribution was biased. The natural follow up was to remove this bias and evaluate the different models, additionally experimenting with other techniques from the PC literature.
> >
> >
> > We hope that our response helped clarify the originality of our contribution and answer the points you have raised about the biased mixture coefficients and normalization. We have expanded the related work section and are currently conducting an experiment with regard to memory capacity.

---

> > > ### Author Response · Authors · 2022-08-05
> > > **Response to Reviewer 7s9W (3)**
> > >
> > > We have included a short experiment regarding capacity in the new revised version. Our results confirm the observation stated in the originial submission that, as you phrase it, "the VAE embeddings are not necessary for strong performance" on the task of recovering patterns corrupted with noise.
> > >
> > > On the task of recovering shifted patterns or patterns with modified colors however, we have shown that adequate embeddings could significantly improve the performance (table 4).
> > >
> > > We hope that this additionnal experiment has answered your concerns, and welcome any further question about our submitted work.

---

> > > > ### Comment · Reviewer_7s9W · 2022-08-08
> > > > **Still problems. Not super happy. But will revise my score to a 5.**
> > > >
> > > > > We agree that this would be a valuable additional experiment, as stated in the discussion about the limitations of our work. Please correct us if we are wrong, but your comment implies that you believe that retrieval scores would deteriorate faster with raw pixel images compared with embeddings, when n increases. What makes you think that ? This prediction is not obvious to us. We are currently running this additional experiment, and we will incorporate the results in the Supplementary Materials.
> > > >
> > > > My main point was that performance would decline with n. This performance decline can be highly non linear and different between models (this is an effect highlighted in https://proceedings.mlr.press/v162/sharma22b.html). If you jointly trained the embedding representations I do think they would do better than the raw pixels simply because the model could learn to embed the data points in more orthogonal ways.
> > > >
> > > > Thanks for running with n>100. But I wish these results had already been collected during the revision period to see how they change the conclusions of the paper.
> > > >
> > > > > We have added this model to the related work, as well as the literature regarding Kanerva Machines and GMM based VAEs, and updated table 1. The comparison with this model might be unfair, as training is necessary in order to store patterns as attractors. In comparison, our models allow us to directly write patterns as columns of the memory matrix.
> > > >
> > > > You are already using and training a VAE that many of the models are sitting on top of. It seems only fair to try the vanilla VAE and see how well it does without any need for convergence or even an explicit storage of the memory matrix. Also given the strong results from that paper I cited it seems all the more important as a baseline.
> > > >
> > > > > If instead we had an entangled representation, not factored according to objects, we suspect that the retrieval rates of the GMM*, NTM* and MEMN2N* models in the last table would be way lower.
> > > >
> > > > I care a lot more about having meaningful vector similarity metrics than I do about lowering the overall performance of every method. I stand by wanting to use another VAE trained from scratch for this dataset.
> > > >
> > > > > In comparison, the Euclidean distance attention does not require any additional trick in order to properly retrieve stored patterns, and is theoretically sound since we derived it from our memory-dependent generative model.
> > > >
> > > > I'd like to see as another baseline L2 normalized vectors that use dot products and thus implement cosine similarity. Your point about LayerNorm not doing this is fair. But it still feels like a confounder to the effects of the different convergence approaches and has been something considered even in newer Transformer variants e.g. https://aclanthology.org/2020.findings-emnlp.379/
> > > >
> > > > > While there is nothing new about attractor dynamics for AMs, using GMMs or relating them to VAEs, this work is the first, to our knowledge, to approach the problem from the PC perspective. We believe that the PC toolbox can bring novel ideas to AM modeling, which can later inspire other advances in this field. Especially, the PC and BP based models we have proposed provide bottom-up information throughout the whole iterative memory retrieval process, which is something that other models are not considering. Additionally, the idea of optimizing the precision matrices (inverse covariance) of the generative model comes from the PC literature. Finally, relating MCHNs to PC is also a novel insight brought by this work.
> > > >
> > > > But can you not find any experiments in which these advantages of PC are not beneficial to performance? In other words why does this relationship to PC matter?
> > > >
> > > > I'll revise my score up to a 5. I am still not super happy with the baselines, n=100 and empirical results. Philosophically, I am not sure if the relations between PC and MCHNs/associative memory are novel enough or profound enough to be worth publication. And as the authors state this is the main contribution rather than empirically increasing performance as the NTM even in the restricted form it is considered here, appears to do better overall.

---

> > > > > ### Author Response · Authors · 2022-08-08
> > > > > **Response to Reviewer 7s9W**
> > > > >
> > > > > Thank you for your comment and for revising your score.
> > > > >
> > > > > > If you jointly trained the embedding representations I do think they would do better than the raw pixels simply because the model could learn to embed the data points in more orthogonal ways.
> > > > >
> > > > > This is a fair point, we did not understand in your first comment that you were suggesting training the VAE as well. In our experiments, the VAE is pretrained in a first stage, and we did not consider this possibility. We agree with you that representations could adapt in order to facilitate memory retrieval. This would constitute another argument in favor of using embeddings instead of raw pixel images for memory retrieval.
> > > > >
> > > > > > You are already using and training a VAE that many of the models are sitting on top of. It seems only fair to try the vanilla VAE and see how well it does without any need for convergence or even an explicit storage of the memory matrix.
> > > > >
> > > > > This is correct. In our case the VAE is not specifically trained on a small set of images that correspond to the patterns in memory, here it is trained on the whole training dataset. We had tried using it as a (weak) baseline at first, but even small perturbations of the input patterns led to extremely low scores, so we stopped experimenting with the vanilla VAE.
> > > > > With additional training on a small set of patterns however, we agree that the vanilla VAE should embed them as attractors and constitute a stronger baseline.
> > > > >
> > > > > > But can you not find any experiments in which these advantages of PC are not beneficial to performance? In other words why does this relationship to PC matter?
> > > > >
> > > > > Right now this relationship with PC does not lead to improved performance, but we believe it brings new insights that could inspire future developments in AM modeling.

---

### Official Review · Reviewer_E281 · 2022-07-15

**Rating:** 5
**Confidence:** 4
**Soundness:** 3 good
**Presentation:** 2 fair
**Contribution:** 2 fair

**Summary:**

The paper attempts to link between ideas in predictive coding (PC) and associative memory (e.g., modern continuous Hopfield network), showing that it is possible to reformulate associative memory retrieval as a kind of variational inference using latent variables. Several inference algorithms are the proposed based on this framework.

**Questions:**

- Please clarify the contributions in the context of existing works (e.g., Kanerva machines and variational memory) as commented in Strengths and Weaknesses.
- The memory M should be introduced with more details. What is the relation between M and M_k in Eq.2? How patterns are stored in the memory?

**Limitations:**

Yes

**Strengths And Weaknesses:**

Strengths
=======
- It is useful to draw connections between separate ideas studied in different communities (variational inference, predictive coding and associative memory).
- The connection between PC and the continuous modern Hopfield network is interesting.

Weaknesses
==========
- The treatment of latent variables conditioned on memory is not entirely new. Several attempts before share similar insights: Kanerva machines [1,2] and variational memory [3].
- The writing is confusing with multiple versions of Eq (2), between framework (Section 3) and methods (Section 4).
- The experiments are somewhat limited.

[1] Wu, Y., Wayne, G., Graves, A., & Lillicrap, T. (2018). The Kanerva Machine: A Generative Distributed Memory. In International Conference on Learning Representations.

[2] Wu, Y., Wayne, G., Gregor, K., & Lillicrap, T. (2018). Learning attractor dynamics for generative memory. Advances in Neural Information Processing Systems, 31.

[3] Le, H., Tran, T., Nguyen, T., & Venkatesh, S. (2018). Variational memory encoder-decoder. Advances in neural information processing systems, 31.

---

> ### Author Response · Authors · 2022-08-02
> **Response to Reviewer E281**
>
> We are grateful for the time you spent on reviewing our work.
>
> Here are our answers to the points you have raised:
>
> Thank you for pointing out the inconsistency in our equation. Indeed, M is not a random variable in our case, so we fixed the notation. We welcome any other comment in order to improve the clarity of our submission.
>
> With regard to our experiments being limited, we have added in the Supplementary Materials a section where we experiment with one-shot generation using the VAE-GMM generative model.
>
> 1. Please clarify the contributions in the context of existing works.
>
> There are indeed other works using latent variables conditioned on memory in the VAE literature. An important limitation of these approaches is that training is needed in order to adapt the amortized inference to a changes in the memory. In comparison, iterative methods such as ours or other attention-based methods directly adapt to the patterns in memory.
>
> Kanerva Machines are more closely related to our approach since they perform iterative inference with a memory based generative model in the variational Bayes framework. On top of the inspiration being different, we can also note several differences with the proposed approaches:
> - in Kanerva Machines, the memory patterns are written in a distributed manner while in our case they are directly added as columns of the memory matrix
> - in Kanerva Machines, the local minima of the energy landscape correspond to stored patterns, while in the PC-GMM and BP-GMM models, the energy function being minimized is constrained by the input (query) pattern. While this is not significant for memory retrieval, this allows to keep some part of the input information if necessary (for instance we could imagine a perceptual inference setting where the memory simply helps inferring some hidden information, and strict convergence to a stored pattern would lose some useful information from the input).
>
> To our knowledge, our contribution is the first work applying PC tools to perform inference in memory-dependent generative models. We have expanded the related work section to include these works as well as other approaches suggested by other reviewers.
>
> 2. The memory M should be introduced with more details. What is the relation between M and M_k in Eq.2? How patterns are stored in the memory?
>
> We have clarified the definition of $M_k$ in the revised manuscript.

---

### Official Review · Reviewer_NGa5 · 2022-07-20

**Rating:** 3
**Confidence:** 3
**Soundness:** 1 poor
**Presentation:** 1 poor
**Contribution:** 1 poor

**Summary:**

The authors explore ways to combine auto-associative memories with variational Bayesian (VB) methods.
  Building on developments in modern Hopfield networks, the authors analyze iterative, memory-dependent inference through the lense of predictive coding.
  Specifically, the authors present:

  1. A VB interpretation of auto-associative memories, and the entailed connection between predictive coding and MCHNs
  2. Four new models that, to a varying degree, combine auto-associative memory and iterative VB inference

**Questions:**

Please, see above.


**Limitations:**

Not applicable.

**Strengths And Weaknesses:**

  Despite the interesting choice of subject matter, I regrettably find the submission to be lacking in many respects.

  1. With respect to the originality, and the scientific context the authors omit several publications.
  More importantly, this omission highlights that the research questions being addressed are poorly motivated.
  I presume that there both practical motives, such as the ability to easily "write" memories, as well fundamental, neuroscientific research questions.
  Unfortunately, the empirical evaluation does not provide such a self apparent justification, and the neuroscientific motivation remains unstated.

  2. The latter point, ties into the clarity of the submission.
  While the writing is clear, I subjectively feel it is somewhat deceptive. Please correct me if I am wrong, but all of the derivations presented in the main text are done under the assumption that the covariances are either fixed or that they are effectively null (e.g Eq. 15 in the appendix).
  For this reason, I do not believe one should frame the problems being addressed nor present the results in any Bayesian fashion.

  3. Given the preceding, I do not believe the presented results offer a particularly valuable contribution. The limited numerical evaluation further solidified this impression. For instance, using the auto-associative memory to do "one-shot learning" subjectively is a natural experiment to conduct. This additionally would present a potential, practical use case for the models introduced.


  Potentially relevant literature:

  1. Ramapuram, J., Wu, Y. & Kalousis, A. Kanerva++: Extending the Kanerva Machine With Differentiable, Locally Block Allocated Latent Memory. in 9th International Conference on Learning Representations, ICLR 2021, Virtual Event, Austria, May 3-7, 2021 (OpenReview.net, 2021).
  2. Wu, Y., Wayne, G., Gregor, K. & Lillicrap, T. Learning Attractor Dynamics for Generative Memory. in Advances in Neural Information Processing Systems vol. 31 (Curran Associates, Inc., 2018).
  3. Wu, Y., Wayne, G., Graves, A. & Lillicrap, T. The Kanerva Machine: A Generative Distributed Memory. in International Conference on Learning Representations (2018).

---

> ### Author Response · Authors · 2022-08-02
> **Response to Reviewer NGa5**
>
> Thank you for the time spent on our submission and for your feedback !
>
> Here are our responses to the points you have raised:
>
> 1. We answer to several remarks in this paragraph:
>
> - "the authors omit several publications"
>
> We have expanded the related work section in order to include other AM approaches such as Kanerva Machines and overparameterized AEs. These approaches are relevant to our work and should not have been omitted to save space. We have also added a short paragraph in the discussion regarding the comparison between PC-inspired models and related work such as Kanerva Machines.
>
> -  "the research questions being addressed are poorly motivated"
>
> Our motivation for this work was to address AM modeling starting from the free-energy principle and PC theory. PC provides solutions for perceptual inference that differ from amortized inference methods such as VAEs. We thought that the PC toolbox could bring novel ideas to AM modeling, which could later inspire other advances in this field. Especially, the PC and BP based models we have proposed provide bottom-up information throughout the whole iterative memory retrieval process, which is something that other models are not considering. Additionally, the idea of optimizing the precision matrices (inverse covariance) of the generative model comes from the PC literature. We hope that this response clarifies our motivation.
>
> 2. "the derivations presented in the main text are done under the assumption that the covariances are either fixed or that they are effectively null. For this reason, I do not believe one should frame the problems being addressed nor present the results in any Bayesian fashion."
>
> In PC, the covariance matrices of the generative model can be optimized, which is usually called "precision weighting". In our case, we have only considered the case of the prior distribution on z and have not optimized the covariance matrices of the intermediate layers. This is a possible avenue of improvement for the model. For simplicity and to have a model that can be used without any training, we have assumed that the covariance matrices were equal to the identity matrix, except for the prior distribution on z in the last proposed model.
>
> If you are referring to the approximate posterior covariance, this is indeed not optimized during PC-based inference. Though, the connection between PC and variational Bayes methods is already well established and is not a claim brought by this work (see for instance [1]).
>
> 3. "using the auto-associative memory to do "one-shot learning" subjectively is a natural experiment to conduct"
>
> We appreciate your recommendation to add a simple experiment that helps evaluating our models. We have added in the Supplementary Materials examples of one-shot generation using our memory-based generative model.
>
>
> We are sorry that you found the soundness and presentation of our work to be poor, and we welcome any suggestion to improve those.
>
> [1] Friston, K., & Kiebel, S. (2009). Predictive coding under the free-energy principle. Philosophical transactions of the Royal Society B: Biological sciences, 364(1521), 1211-1221.

---

### Author Response · Authors · 2022-08-08
**Response to all reviewers**

We would like to thank all the reviewers for the time spent reviewing our paper. We believe that we have addressed all the concerns and questions raised in the reviews so far. Since the end of the rebuttal is approaching, please indicate if there are any other remaining concerns, question or suggestion, we would be pleased to address those points. If you are satisfied with our response, we would appreciate it if you would update your score to account for the changes we have made to the manuscript.

Your feedbacks have greatly helped improving the clarity in the models and positioning of the article, here is the summary of the updates made to the manuscript:

- Improvement of the related work section which now includes and discusses more related associative memory models such as Kanerva Machines, overparameterized auto-encoders, or Gaussian mixtures VAEs (section 2.1)
- Additional results experimenting with one-shot generation (section E.3)
- Additional results evaluating the capacity of the proposed models with larger memory stores (section E.4)
- Clarification of the BP-GMM algorithm, and less focus on the PC-GMM algorithm, as suggested by reviewers (algorithm 2)
- New figure summarizing the different iterative approaches we propose (figure 1)

Additionally, in our response we have clarified some slight misunderstandings regarding the methods (biased GMM vs balanced GMM) and the originality of our work (first work using PC inspired techniques in a variational inference setting with memory-dependent generative models).

---

### Public Comment · ~Jerry_Yao-Chieh_Hu1 · 2023-08-13
**Why Should We Assume the Approximate Posterior Is Tightly Shaped Around Its Mean?**

Dear Authors,

In the General VFE derivation (Sec. A of appendix), you assume the approximate posterior is tightly shape around its mean $\hat{z}$. Can you confirm whether this assumption holds universally?

I may have overlooked something, but I was unable to fully grasp the rationale for this assumption. Could you provide further explanation or perhaps point me to relevant references to aid my understanding?

Thank you!

---

> ### Public Comment · ~Louis_Annabi1 · 2023-08-14
> **Reponse to Jerry Yao-Chieh Hu**
>
> Dear Jerry,
>
> I would recommend looking at the mathematical derivations presented in ([Buckley et al., 2017](https://www.sciencedirect.com/science/article/pii/S0022249617300962)), in particular section 3. and 4., and the discussion about the assumptions that were made.
>
> This assumption means that the variance (or covariance) of the approximate posterior is small. I do not see any justification for this assumption other than the fact that it simplifies the expression of the VFE and eventually leads to equations closely resembling PC. I interpret this assumption as an approximation made by the PC algorithm, that focuses only on optimizing the approximate posterior mean. We can still represent and optimize uncertainties about $z$ in the generative model $p(z)$.
>
> I hope this helps,
>
> Louis Annabi

---

### Meta-Review · Area_Chair_E6mp · 2022-08-24

**Recommendation:** Accept
**Confidence:** Less certain

**Metareview:**

This submission provides a link between auto-associative memory and variational inference by making priors of representations being memory dependent. This view provides new important insights, as well as new algorithms.

The reviewers raised concerns mostly about 1) the clarity of the presentation, and 2) connections to previous work. The authors addressed both of these points during the rebuttal stage. Still some questions remain about performance evaluation, but the AC is of the opinion that these concerns are minor compared to the conceptual advances the paper makes.



**Award:**

No

---

### Decision · Program_Chairs · 2022-09-14

Accept